# LEDiT: Your Length-Extrapolatable Diffusion Transformer without Positional Encoding

**Shen Zhang**[1]  **Siyuan Liang**[1]  **Yaning Tan**[2]  **Zhaowei Chen**[1]  **Linze Li**[1]  **Ge Wu**[3]
**Yuhao Chen**[1]  **Shuheng Li**[1]  **Zhenyu Zhao**[1]  **Caihua Chen**[2]  **Jiajun Liang**[1*]  **Yao Tang**[1*]
[1]JIIOV Technology    [2]Nanjing University    [3]Nankai University
shen.zhang@jiiov.com   tracyliang18@gmail.com   yao.tang@jiiov.com

## Abstract

Diffusion transformers (DiTs) struggle to generate images at resolutions higher than their training resolutions. The primary obstacle is that the explicit positional encodings (PE), such as RoPE, need extrapolating to unseen positions which degrades performance when the inference resolution differs from training. In this paper, We propose a Length-Extrapolatable Diffusion Transformer (LEDiT) to overcome this limitation. LEDiT needs no explicit PEs, thereby avoiding PE extrapolation. The key innovation of LEDiT lies in the use of causal attention. We demonstrate that causal attention can implicitly encode global positional information and show that such information facilitates extrapolation. We further introduce a locality enhancement module, which captures fine-grained local information to complement the global coarse-grained position information encoded by causal attention. Experimental results on both conditional and text-to-image generation tasks demonstrate that LEDiT supports up to 4× resolution scaling (e.g., from 256×256 to 512×512), achieving better image quality compared to the state-of-the-art length extrapolation methods. We believe that LEDiT marks a departure from the standard RoPE-based methods and offers a promising insight into length extrapolation. Project page: https://shenzhang2145.github.io/ledit/

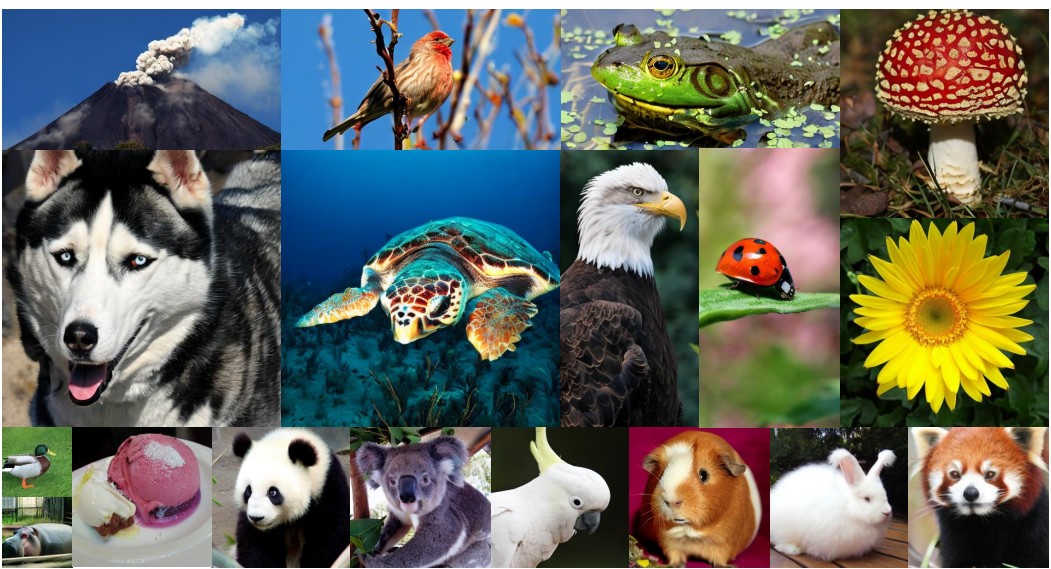

Figure 1: Selected arbitrary-resolution samples ($512^2$, $512\times256$, $256\times512$, $384^2$, $256^2$, $128^2$) from LEDiT-XL/2 trained on ImageNet $256\times256$ resolution. LEDiT can generate high-quality images beyond the limitations of training resolution.

---

[*]Corresponding author.

39th Conference on Neural Information Processing Systems (NeurIPS 2025).

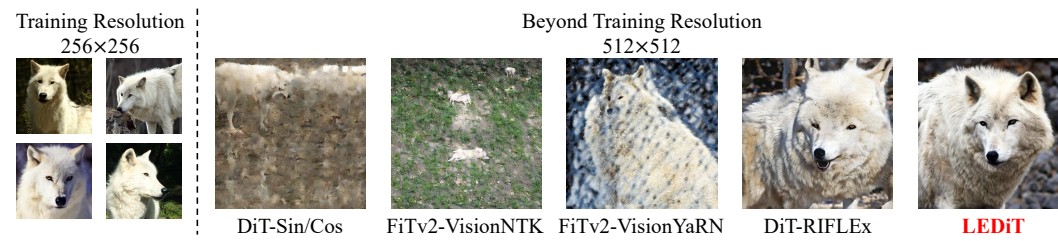

| Training Resolution 256×256 | | Beyond Training Resolution 512×512 | | | |
| DiT-Sin/Cos | FiTv2-VisionNTK | FiTv2-VisionYaRN | DiT-RIFLEx | **LEDiT** |

Figure 2: Diffusion Transformer performs well at the training resolution. However, when extrapolated to higher resolutions, DiT [31], FiT [28, 43], and RIFLEx [50] suffer notable quality degradation. In contrast, our LEDiT can generate reasonable and realistic higher-resolution images with fine-grained details. The class label is 270 (white wolf).

# 1 Introduction

Diffusion models have emerged as a powerful foundation technique in vision generation tasks. The architecture of diffusion models has progressed from U-Net [36, 35] to transformer-based designs [31, 1]. Diffusion Transformers (DiTs) have become state-of-the-art generators [11, 4, 48, 44]. Despite their success, DiTs face critical limitations when generating images at resolutions beyond those encountered during training [28, 42]. As shown in Figure 2, DiTs trained on ImageNet [6] 256×256 resolution produce high-quality samples at this scale, but struggle to generalize to higher resolutions such as 512×512. Due to the expensive quadratic cost of self-attention and the scarcity of large-scale, high-resolution datasets, models are typically trained at relatively small resolutions. In practice, many real-world applications, such as high-definition film and computer graphics, require higher-resolution images, presenting a significant length extrapolation challenge for current DiTs.

Many studies [33, 39, 32, 23, 28] highlight the importance of positional encoding (PE) in length extrapolation. Rotary Positional Embeddings (RoPE) [39] and its variants, such as NTK-aware scaling [2] and YaRN [32] have been developed to improve the extrapolation ability of language transformers. In the vision domain, Flexible Vision Transformers (FiT) [28, 43] integrate RoPE into DiTs to support variable input resolutions. RIFLEx [50] reduced the intrinsic frequency of RoPE to alleviate extrapolation issues. Despite these advances, performance still degrades notably beyond the training resolution (see Figure 2). Since diffusion models are not trained for such out-of-range positions, this leads to a distribution shift in positional indices and results in out-of-distribution issues [8]. On the other hand, Recent studies [15, 23, 5] challenge the need of explicit PE, showing that large language models (LLMs) without PE (NoPE) perform well in in-distribution settings and even outperform explicit PEs in length extrapolation. The advantage of NoPE lies in avoiding PE extrapolation, which reduces performance when inference resolution differs from training. LookHere [12] removes the positional encoding and carefully designs various combinations of causal masks, achieving notable success in image recognition. However, directly applying LookHere to image generation tasks results in severe object duplication (see Figure 15) and fails to yield effective length extrapolation. Therefore, it remains unclear whether DiTs can similarly benefit from NoPE for resolution extrapolation. We ask: **Can DiTs leverage NoPE to train at low resolutions and generalize to higher resolutions?**

In this paper, We propose a **Length-Extrapolatable Diffusion Transformer** (LEDiT), which removes explicit PE and can generate high-quality images at arbitrary resolutions. A key architectural modification is the adoption of causal attention. We demonstrate that causal attention can implicitly encode positional information. Specifically, we provide both theoretical and empirical evidence that token variance decreases when position increases, providing an implicit ordering (see Section 3.2). We reveal that this implicit position ordering yields better extrapolation abilities than explicit PEs. Furthermore, we introduce a negligible-cost multi-dilation convolution as a locality enhancement module to improve local fine-grained details, complementing the global coarse-grained information captured by causal attention.

We conduct extensive experiments on both conditional and text-to-image generation tasks to validate the effectiveness of LEDiT. Notably, LEDiT supports up to 4× inference resolution scaling while maintaining structural fidelity and fine-grained details, outperforming state-of-the-art extrapolation methods. Moreover, LEDiT can generate images with arbitrary aspect ratios (e.g., 512×384 or 512×256) without any multi-aspect-ratio training techniques. We also show that fine-tuning LEDiT

from a pretrained DiT for only 100K steps yields strong extrapolation performance, highlighting its potential for efficient integration into existing powerful DiTs. We hope our findings provide valuable insights for future research on transformer length extrapolation.

## 2  Related Work

**Diffusion Transformers.** Building on the success of DiT [31, 1], subsequent works such as PixArt-Alpha [4] and PixArt-Sigma [3] further extend diffusion transformers for higher-quality image generation. Stable Diffusion 3 [11] and Flux [25] substantially improve the performance of diffusion transformers by scaling up parameters. Sana [46] focused on fast generation through deep compression autoencoding and linear attention Despite these advances, most DiTs struggle when inference resolution differs from training, motivating our exploration of length extrapolation.

**Length Extrapolation in Language.** Since the introduction of the transformer [41], length extrapolation has remained a significant challenge, with positional encoding playing a critical role [33, 39, 32, 23, 29]. Absolute Positional Encoding (APE) [41] struggles to handle longer sequences. To address this, ALiBi [33] modifies attention biases to facilitate length extrapolation. Rotary Position Embedding (RoPE) [39] and its extrapolation refinements, including NTK-aware scaling [2] and YaRN [32], further improve length generalization. Adaptive embedding schemes like Data-Adaptive Positional Encoding (DAPE) [51] and Contextual Positional Encoding (CoPE) [13] have also been explored. In contrast to explicit PEs, NoPE [5, 23] demonstrates that language models can implicitly encode positional information. We independently observe a similar phenomenon in diffusion models. Importantly, compared to [5], our analysis provides a theoretical proof under more relaxed assumptions, requiring only finite mean and variance and imposing weaker constraints on weight matrices. Furthermore, we demonstrate that this implicit positional information benefits length extrapolation in diffusion models, which is the main focus of this paper.

**Length Extrapolation in Diffusion.** Length extrapolation has been extensively studied in diffusion U-Net architectures [16, 49, 10, 20, 14], but remains largely unexplored in DiTs. RoPE-Mixed [17] employs rotation-based embeddings for variable image sizes. FiT [28, 43] adopts RoPE, NTK-Aware, and YaRN in 2D variants for resolution extrapolation. RIFLEx [50] analyzes the role of different frequency components in RoPE and found that reducing the intrinsic frequency can boost length extrapolation. LookHere [12] carefully designs various combinations of causal masks to provide directional inductive biases. It conducts experiments to demonstrate the extrapolation capabilities in classification tasks. However, directly adapting LookHere to image generation tasks results in severe object duplication (see Figure 15). Therefore, a comprehensive strategy for high-resolution extrapolation in diffusion transformers remains elusive. In this work, we address this gap by enabling DiTs to generate high-fidelity images at arbitrary resolutions. There are some conceptual similarities between LEDiT and LookHere. Both LEDiT and LookHere explore causal attention to enhance length extrapolation. But there are key differences. LEDiT provides a theoretical framework that explains why causal attention is capable of encoding positional information and enabling length extrapolation. Moreover, LEDiT combines simple causal attention with multi-dilation convolutions, effectively mitigating object duplication (see Figure 15) and achieving better extrapolation performance.

## 3  Method

We first introduce some preliminaries about DiTs and causal attention. DiTs is primarily built upon the ViT [9]. Each DiT block contains a multi-head self-attention (MSA), followed by adaptive layer normalization (AdaLN) and a feed-forward network (MLP). Residual connections are applied by scaling $\alpha_\ell$ and $\alpha'_\ell$. Given an input $x \in \mathbb{R}^{H \times W \times C}$, the computation of DiT block is as follows:

$$z_0 = \text{Flatten}(\text{Patchify}(x)) + E_{\text{pos}}, \tag{1}$$

$$z'_\ell = \text{MSA}(\text{adaLN}(z_{\ell-1}, t, c)) + \alpha_\ell z_{\ell-1}, \tag{2}$$

$$z_\ell = \text{MLP}(\text{adaLN}(z'_\ell, t, c)) + \alpha'_\ell z'_\ell. \tag{3}$$

Causal attention only allows the given position in a sequence to attend to the previous positions, not to future positions. The causal attention map is:

$$A = \text{softmax}\left(\frac{QK^\top}{\sqrt{d_k}} + M\right), \tag{4}$$

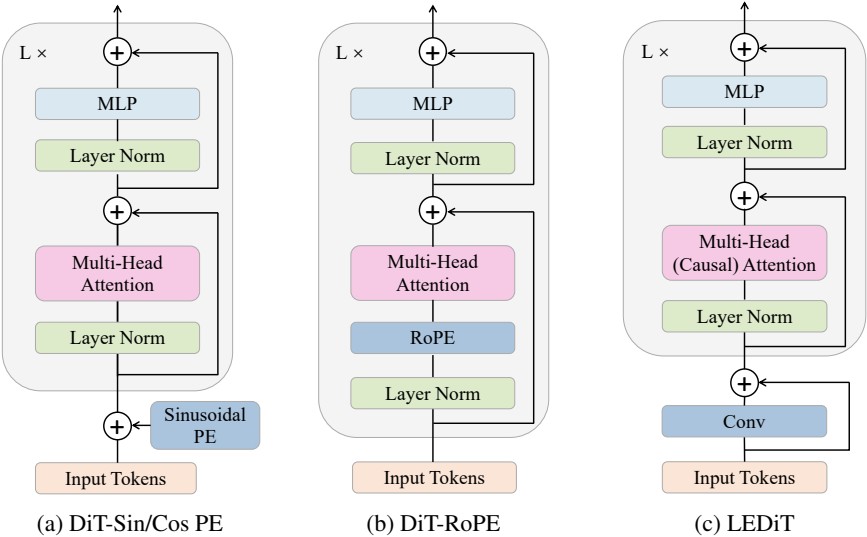

| (a) DiT-Sin/Cos PE | (b) DiT-RoPE | (c) LEDiT |

Figure 3: Comparison between DiT-Sin/Cos PE, DiT-RoPE, and our LEDiT. We omit AdaLN for the sake of simplicity. DiT-Sin/Cos PE is the vanilla DiT [31], which incorporates Sinusoidal PE into the transformer. DiT-RoPE introduces rotary position encoding by rotating the query and key in each transformer block. In contrast, our LEDiT model does not require explicit position encoding. The main difference lies in the incorporation of causal attention and convolution after patchification.

where $Q \in \mathbb{R}^{n \times d_k}$ and $K \in \mathbb{R}^{n \times d_k}$ are query and key, $d_k$ is the dimension, and $M \in \mathbb{R}^{n \times n}$ is a mask matrix with definition as follows:

$$M_{i,j} = \begin{cases} 0 & \text{if } j \leq i, \\ -\infty & \text{if } j > i. \end{cases} \tag{5}$$

This ensures attention scores for future tokens are nearly zero after softmax, enforcing strict causality.

## 3.1 LEDiT Block

The overall architecture of LEDiT is illustrated in Figure 3c. Our LEDiT does not need explicit PEs. The main modifications include the use of causal attention and a negligible-cost multi-dilation convolution. We design LEDiT blocks to alternate between causal attention and self-attention. The first LEDiT block uses self-attention, formulated as:

$$z'_\ell = \text{MSA}(\text{adaLN}(z_{\ell-1}, t, c)) + \alpha_\ell z_{\ell-1}, \tag{6}$$

$$z_\ell = \text{MLP}(\text{adaLN}(z'_\ell, t, c)) + \alpha'_\ell z'_\ell. \tag{7}$$

The subsequent LEDiT block uses causal attention, which can be written as:

$$z'_{\ell+1} = \text{MCA}(\text{adaLN}(z_\ell, t, c)) + \alpha_{\ell+1} z_\ell, \tag{8}$$

$$z_{\ell+1} = \text{MLP}(\text{adaLN}(z'_{\ell+1}, t, c)) + \alpha'_{\ell+1} z'_{\ell+1}, \tag{9}$$

where MCA represents multi-head causal attention. We explore more LEDiT designs in Table 1d.

## 3.2 Why Causal Attention

Explicit PEs are widely used in transformers, but their performance degrades when extrapolating to resolutions larger than those seen during training, as shown in Figure 5. To address this limitation, we attempt to remove explicit PEs to avoid position encoding extrapolation. Prior work [23] suggests that causal attention enables better length extrapolation without PEs in LLM. Motivated by this, we introduce causal attention to diffusion models and further demonstrate that causal attention (i) implicitly encodes positional information to tokens, and (ii) that such implicit positional encodings facilitate length extrapolation.

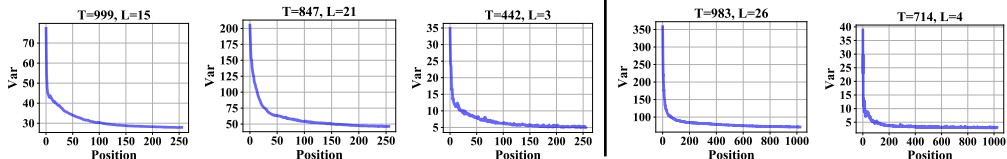

Figure 4: $\text{Var}(y_{il})$ distributions across various timestep (T) and DiT layers (L). Left: variance distribution at the training resolution (256×256). Right: variance distribution beyond the training resolution (512×512). Best viewed when zoomed in.

**Causal attention implicitly encodes positional information.** we formally establish that, under specific assumptions, the variance of causal attention output encodes positional information. Specifically, we prove the following theorem:

**Theorem 3.1.** *For a Transformer architecture with Causal Attention, assume that the value $V$ is i.i.d. with mean $\mu_V$ and variance $\sigma_V^2$. Then, the variance of the causal attention output $Y_{il}$ at position $i$ and dimention $l$ is given by:*

$$\text{Var}(Y_{il}) = \frac{2}{i+1}\,\sigma_V^2 + \frac{i-1}{i(i+1)}\,\mu_V^2. \tag{10}$$

*When $i$ is large, we can approximate $\frac{i-1}{i(i+1)} \approx \frac{1}{i+1}$, leading to the reasonable approximation*

$$\text{Var}(Y_{il}) \approx \frac{C}{i+1}, \tag{11}$$

*where the constant $C = 2\sigma_V^2 + \mu_V^2$.*

Please refer to Appendix A for the complete proof. This theorem reveals that, if the conditions are met, the variance is inversely proportional to the position $i$ at a rate of $\frac{1}{i+1}$. We further conduct experiments to verify whether applying causal attention in DiT can assign different variances to different positions.

We train a DiT-XL/2 that replaces all self-attention with causal attention and use it to verify the theorem. Given an input sequence $z \in \mathbb{R}^{n \times d_k}$, causal attention takes $z$ and outputs $y = (y_1, ..., y_n) \in \mathbb{R}^{n \times d_k}$. We approximate $\text{Var}(y_{il})$ using the variance of $y_i$. As shown in the left figures of Figure 4, $\text{Var}(y_{il})$ is inversely proportional to

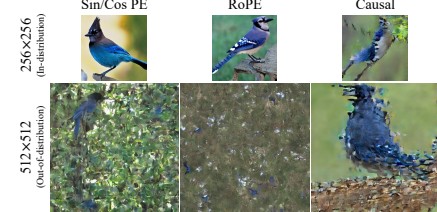

Figure 5: Explicit PEs degrade in extrapolation, while causal attention consistently outputs coherent coarse-grained structures. The class label is 17 (jay).

the position $i$ across various timestep and layers. This indicates the existence of causal attention in DiT that meets the conditions of the theorem. Intuitively, a smaller $\text{Var}(y_i)$ indicates that its elements are more concentrated. During training, the neural network can learn to leverage this concentration to determine token position, thereby implicitly encoding positional ordering. We also observe variance distribution in the later denoising stage differs from the theorem. To evaluate the impact of this phenomenon, we conduct experiments with switching from causal attention to self-attention at different timesteps. Our findings show that positional information is primarily acquired in the early denoising stage, and the causal attention with variance deviation in the later denoising stage has minor effects on extrapolation. See Section C for the discussion.

**The implicit positional information facilitates length extrapolation.** As shown in the right figures of Figure 4, when extrapolating to higher resolutions, the variance remains inversely proportional to the position, consistent with the in-distribution variance distribution. This preservation of implicit positional ordering enables the model to generalize across larger resolutions, as illustrated in Figure 5. When scaling the resolution by 4×, models with explicit positional encodings exhibit severe structural degradation, while models with causal attention continue to generate structurally coherent objects. In addition to providing implicit positional information to tokens, causal attention also acts as a learnable, global receptive-field mechanism, which differs from static positional encodings. It can make predictions based on previous tokens and learn the dependencies between tokens from large-scale data, which may also enhance the model's ability to extrapolate to higher resolution.

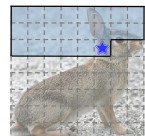 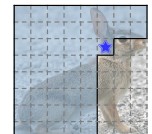 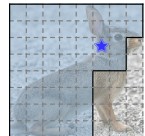 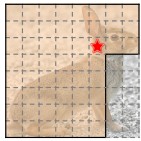

(a) One-Dimension     (b) Mask Lower-right     (c) Unmask Neighborhood     (d) Mask Lower-right Corner

Figure 6: Comparison of causal attention scan variants. We use variant (d) as our default.

**Causal scan variants.** We introduce four causal attention scan variants, as depicted in Figure 6 (a) represents the traditional 1D scan used in PixelCNN [40], where each position attends only to preceding tokens in a flattened sequence. To leverage the spatial characteristics of images, we also consider scanning along both the height and width dimensions and propose (b)–(d). We ablate the performance of these variants in Table 1b. Variant d is set as our default.

### 3.3 Locality Enhancement

Although causal attention provides tokens with global implicit positional ordering, when $i$ is large, the variance between adjacent tokens becomes indistinguishable (see Figure 4), preventing accurate position information and leading to blurry images, see Figures 5 and 18a. To distinguish the relative relationships between neighborhood tokens, we need to enhance the local perception abilities of the neural network. Specifically, we introduce convolution as a locality enhancement module. Previous work [45] replaced the qkv-projector with convolution or integrated convolution into the MLP [47, 46] in each transformer block, which significantly increased the model's computational cost. We find that adding a convolution after patchification is sufficient, while only increasing ignorable overhead. This can be written by slightly modifying Equation (1):

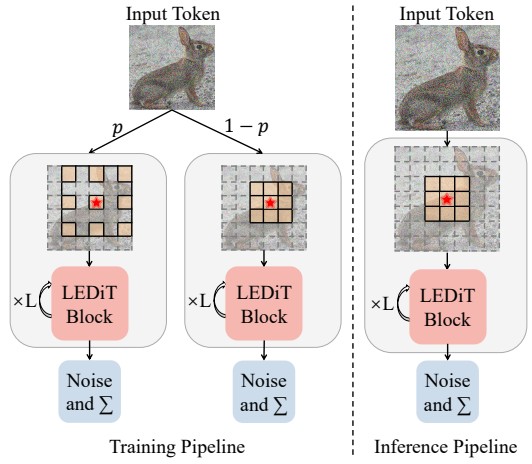

Figure 7: LEDiT pipeline.

$$z_0 = \text{Flatten}(C_{3,1,1,1}(\text{Patchify}(x))), \tag{12}$$

where $C_{k,p,s,d}$ denotes a convolution filter with kernel size $k$, padding $p$, stride $s$, and dilation $d$. Zero padding is applied, which enables convolution to leak local positional information [21, 47].

**Multi-dilation training strategy.** Although the generated higher-resolution images are visually compelling, they often encounter duplicated object artifacts due to the fixed receptive fields of convolutional kernels [16]. To mitigate this problem, we adopt a multi-dilation training strategy, wherein dilation and padding are randomly adjusted during training (see Figure 7). For a standard convolution filter $C_{3,1,1,1}$, we set a probability $p$ to expand both its dilation rate and padding size to 2, transforming it into $C_{3,2,1,2}$. During inference, we empirically find that fixing dilation and padding as 1 is sufficient. This strategy trains shared-parameter convolutions with varying receptive fields and empirically improves extrapolation abilities.

## 4 Experiments

### 4.1 Experiment Settings

**Model Architecture.** For conditional generation on ImageNet [6], we use a patch size $p = 2$ and follow DiT-XL [31] to set the same layers, hidden size, and attention heads for the XLarge model, denoted by LEDiT-XL/2. For text-to-image generation on COCO [26], We use MMDiT [11] and set the hidden dimension as 768 and the model depth as 24, following the design in REPA [48], denoted as LEMMDiT. We use the CLIP [34] text encoder to compute text captions.

Table 1: Ablations using LEDiT-XL/2 on 256×256 ImageNet. We report FID and IS scores. For each ablation, we load the pretrained DiT weights and fine-tune LEDiT-XL/2 for 100K iterations. Default settings are marked in gray . See Figure 18 for visualization.

(a) **Components ablations.** Causal attention and convolution are effective in length extrapolation.

| case | FID↓ | IS↑ |
|---|---|---|
| NoPE | 378.95 | 3.79 |
| + Cau. | 286.01 | 6.96 |
| + Con. | 130.91 | 28.66 |
| + Cau. + Con. | **35.86** | **139.91** |

(b) **Casual scan variants.** 2D casual scan variants outperform 1D variants.

| scan | FID↓ | IS↑ |
|---|---|---|
| (a) | 62.49 | 78.25 |
| (b) | 89.77 | 50.10 |
| (c) | 43.17 | 116.03 |
| (d) | **35.86** | **139.91** |

(c) **Multi-dilation strategy.** LEDiT benefits from multi-dilation strategy.

| case | FID↓ | IS↑ |
|---|---|---|
| w/o multi-dila | 39.20 | 127.84 |
| w/ multi-dila | **35.86** | **139.91** |

(d) **Block design**. The alternating order works better than the sequential order.

| order | FID↓ | IS↑ |
|---|---|---|
| $CA_{L/2} + SA_{L/2}$ | 36.81 | 139.88 |
| $SA_{L/2} + CA_{L/2}$ | 48.65 | 103.14 |
| $(CA,SA)_{L/2}$ | 36.05 | **143.26** |
| $(SA,CA)_{L/2}$ | **35.86** | 139.91 |

(e) **Multi-dilation probability**. LEDiT with a small probability works better.

| prob | FID↓ | IS↑ |
|---|---|---|
| 0 | 39.20 | 127.84 |
| 0.1 | **35.86** | **139.91** |
| 0.2 | 37.99 | 135.85 |
| 0.5 | 37.56 | 133.07 |

(f) **Dilation rate**. (2,3) means randomly selecting 2 or 3 as the rate during training.

| dilation | FID↓ | IS↑ |
|---|---|---|
| 1 | 39.20 | 127.84 |
| 2 | **35.86** | **139.91** |
| (2,3) | 37.24 | 136.23 |

**Training Details.** The experiments are trained on ImageNet [6] with 256×256 and 512×512 resolutions, and on COCO [26] with 256×256 resolution. On ImageNet, We (i) train the randomly initialized LEDiT for 400K steps or (ii) fine-tune LEDiT for 100K steps. We set the batch size as 256. On COCO, We follow REPA [48] and train LEMMDiT for 200K steps with a batch size of 192. We use 8× NVIDIA V100 GPUs as default training hardware.

**Evaluation Metrics.** We primarily use Fréchet Inception Distance (FID) [18], the standard metric for evaluating generative models. We additionally report Inception Score [37], sFID [30], and Precision/Recall [24] as secondary metrics. Without further elaboration, on ImageNet, we generate 50K samples using 250 DDPM sampling steps with a classifier-free guidance (CFG) scale of 1.5. On COCO, we generate 40,504 images (one per caption) using 50 ODE sampling steps with CFG=2.0. For fair comparison, all values reported in this paper are obtained by exporting samples and using ADM's TensorFlow evaluation suite [7].

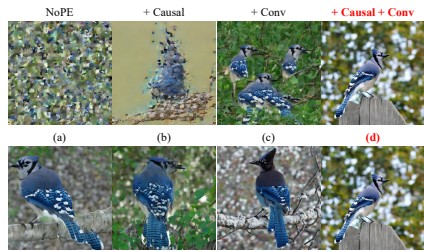

Figure 8: Visualization of the ablation study. The first row illustrates the ablations of the components proposed in this paper, while the second row displays the ablations of the causal scan variants. The models are trained on 256× 256 ImageNet and generate images with 512×512 resolution. See Figure 18 for more visualization.

**Evaluation Resolution.** Compared to previous work [28, 43], this paper tests at more extreme resolutions. When trained on ImageNet 256×256, we compare the extrapolation performance of LEDiT with other methods [31, 28, 43, 50] at 384×384 (2.25×), 448×448 (about 3×), and 512×512 (4×) resolutions. When trained on ImageNet 512×512, we compare LEDiT with other methods at 768×768 (2.25×), 896×896 (about 3×), and 1024×1024 (4×) resolutions. Additionally, we assess performance at different aspect ratios, specifically 512×384 (3:2) and 384×512 (2:3). On COCO, extrapolation is evaluated at 512×512 (4×). All token lengths are much longer than those seen during training. Following the widely adopted practice in transformers, we apply attention scaling [22] for length extrapolation.

## 4.2 LEDiT Ablations

In this section, we ablate LEDiT design settings on 256×256 ImageNet. We use LEDiT-XL/2 to ensure that our method works at scale. We evaluate performance by loading DiT pretrained weights

Table 2: Comparison of state-of-the-art extrapolation methods and LEDiT trained on 256×256 ImageNet at various resolutions beyond the training image size. We set CFG=1.5. * indicates training from scratch. † indicates additional architecture refinement.

| Model | 384×384 | | | | | 448×448 | | | | | 512×512 | | | | |
|---|---|---|---|---|---|---|---|---|---|---|---|---|---|---|---|
| | FID↓ | sFID↓ | IS↑ | Prec.↑ | Rec.↑ | FID↓ | sFID↓ | IS↑ | Prec.↑ | Rec.↑ | FID↓ | sFID↓ | IS↑ | Prec.↑ | Rec.↑ |
| DiT-Sin/Cos PE* | 114.10 | 162.50 | 14.91 | 0.18 | 0.27 | 188.42 | 191.58 | 4.19 | 0.06 | 0.11 | 216.22 | 188.69 | 2.70 | 0.10 | 0.04 |
| DiT-VisionNTK* | 45.81 | 80.42 | 99.92 | 0.48 | 0.42 | 124.88 | 113.88 | 37.79 | 0.22 | **0.39** | 174.68 | 139.23 | 16.28 | 0.10 | 0.30 |
| DiT-VisionYaRN* | 23.45 | 53.25 | 138.46 | 0.63 | 0.35 | 64.93 | 88.59 | 70.04 | 0.36 | 0.34 | 109.00 | 109.88 | 38.38 | 0.21 | 0.30 |
| DiT-RIFLEx* | 18.47 | 64.36 | 156.34 | 0.66 | **0.38** | 49.29 | 92.25 | 81.78 | 0.42 | 0.35 | 119.57 | 107.32 | 29.44 | 0.17 | 0.30 |
| LEDiT* | 15.98 | 30.94 | 138.25 | 0.75 | 0.31 | 29.84 | **48.06** | 103.05 | 0.61 | 0.25 | 56.02 | 65.99 | 63.26 | 0.43 | 0.21 |
| LEDiT*† | **12.07** | **30.47** | **188.15** | **0.80** | 0.31 | **20.91** | 48.37 | **152.57** | **0.69** | 0.25 | **34.29** | **64.10** | **110.04** | **0.56** | 0.22 |
| DiT-Sin/Cos PE | 87.03 | 116.67 | 44.93 | 0.31 | 0.31 | 168.23 | 145.45 | 15.25 | 0.12 | 0.23 | 213.77 | 168.51 | 7.98 | 0.06 | 0.13 |
| DiT-VisionNTK | 71.23 | 80.69 | 67.42 | 0.33 | 0.51 | 184.29 | 122.99 | 16.94 | 0.10 | 0.41 | 246.56 | 144.99 | 8.82 | 0.04 | 0.17 |
| DiT-VisionYaRN | 13.51 | 35.35 | 244.42 | 0.71 | 0.39 | 28.23 | 50.81 | 170.22 | 0.56 | 0.35 | 49.86 | 64.63 | 109.34 | 0.42 | **0.35** |
| DiT-RIFLEx | 57.88 | 77.27 | 75.88 | 0.37 | **0.55** | 186.76 | 129.17 | 17.13 | 0.09 | 0.41 | 251.92 | 163.77 | 10.39 | 0.04 | 0.12 |
| FiTv2-VisionNTK | 38.43 | 47.09 | 107.89 | 0.45 | 0.54 | 179.01 | 117.12 | 18.20 | 0.08 | 0.42 | 257.63 | 171.10 | 6.72 | 0.01 | 0.21 |
| FiTv2-VisionYaRN | 23.23 | 35.13 | 157.93 | 0.55 | 0.48 | 71.94 | 64.72 | 64.49 | 0.29 | **0.51** | 155.80 | 118.21 | 20.76 | 0.11 | 0.27 |
| LEDiT | **9.34** | **25.02** | **281.09** | **0.78** | 0.39 | **17.62** | **39.43** | **214.90** | **0.66** | 0.34 | **33.25** | **54.36** | **138.01** | **0.52** | 0.31 |

and fine-tuning for 100K iterations. we set CFG=1.5, generate 10K images at 512×512 resolution, and report FID-10K and IS-10K.

**Components ablations.** Table 1a shows the influence of each component of LEDiT. Removing PE (NoPE) degrades DiT severely. Both causal attention and convolution can significantly enhance extrapolation performance. Combining these two components decreases the FID from 378.95 to 35.86 and increases the IS from 3.79 to 139.91, yielding the optimal performance. Figure 8 illustrates the impact of causal attention and convolution. DiT with NoPE generates noise-like images, indicating that without PE, DiT cannot capture token positional information. Incorporating causal attention yields structurally coherent objects but insufficient high-frequency details, since causal attention provides a global ordering but struggles with local distinctions (see Section 3.3). Conversely, introducing convolution provides adequate high-frequency details, but leads to duplicated objects due to the lack of a global receptive field. When both are combined, the generated images exhibit realistic object structures as well as fine-grained details. Therefore, we use both causal attention and convolution as the default setting.

**Causal scan variants.** Table 1b presents a quantitative comparison of different scan variants. Except for variant (b), 2D scan variants (c) and (d) outperform the 1D variant (a) in both FID and IS. Figure 8 shows that variants (c) and (d) produce more coherent object structures and finer details than variant (a). Although variant (b) is also a 2D scan, it leads to blurred images, resulting in lower FID and sFID scores. We adopt variant (d) as our default.

**Multi-dilation strategy.** As shown in Figure 18c, when adapting to multiple receptive fields, the multi-dilation strategy enhances LEDiT's performance and significantly mitigates object duplication. Table 1c shows that LEDiT with the multi-dilation strategy achieves better FID and IS scores. We adopt the multi-dilation strategy as the default setting.

**Block design.** Table 1d compares different orders of causal and self-attention blocks. The first two rows use sequential blocks, whereas the last two employ an alternating arrangement. Both orders achieve strong performance, but sequential order exhibits higher variance while alternating orders are more stable. We thus use the alternating order with self-attention preceding causal attention as our default.

**Multi-dilation probability.** Table 1e shows the result of different multi-dilation probabilities $p$, where $p = 0$ disables the strategy. As $p$ increases, FID initially decreases and then rises. At $p = 0.5$, the receptive field alternates equally between 3×3 and 5×5. The experimental results show that frequent conv parameter changes during training slow convergence and introduce instability, leading to worse performance, while smaller $p$ mitigates object duplication and ensures stable training. So we adopt $p = 0.1$ as the default setting.

**Dilation rate.** We also evaluate different dilation rates $r$ to accommodate multiple receptive fields (Table 1f). For instance, $r = (2, 3)$ means there is a $p/2$ probability of choosing $r = 2$ or $r = 3$. It can be seen that both $r = 2$ and $r = (2, 3)$ can improve performance, but $r = (2, 3)$ is less effective than $r = 2$, likely due to the increased complexity of handling multiple dilation values. Nevertheless, successfully adapting convolution to multiple receptive fields may further benefit extrapolation. In this paper, we retain $r = 1$ by default.

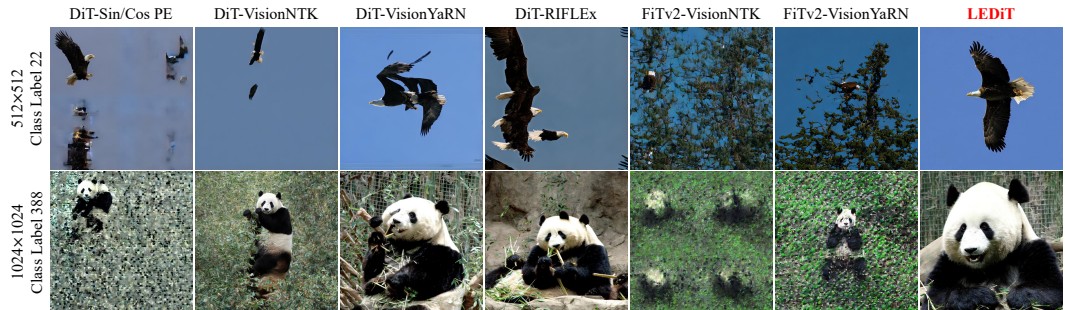

Figure 9: Qualitative comparison with other methods. The resolution and class label are located to the left of the image. We use the model trained on 256×256 ImageNet to generate images at 512×512 resolution, and the model trained on 512×512 ImageNet to generate images at 1024×1024 resolution. We set CFG=4.0. Best viewed when zoomed in. See Figure 19 and Figure 20 for more comparison.

## 4.3 Main Results

**256×256 ImageNet.** In Figure 9, we present a qualitative comparison between LEDiT and other methods. Vanilla DiT (DiT-Sin/Cos PE) suffers from severe image quality degradation. When combined with VisionNTK, VisionYaRN, or RIFLEx, DiT generates images with detailed textures but introduces unrealistic object structures. FiT produces images with severely degraded quality. In contrast, LEDiT produces images with realistic object structures and rich details. The quantitative comparison is reported in Table 2. LEDiT substantially outperforms previous extrapolation methods. At a resolution of 384×384, LEDiT reduces the previous best FID-50K of 13.51 (achieved by DiT-VisionYaRN) to 9.34. As the resolution increases, LEDiT further widens the performance gap, lowering the best previous score from 49.86 to 33.25 at 512×512. When trained from scratch, LEDiT also achieves significantly better FID and sFID scores compared to its counterparts, demonstrating the effectiveness of LEDiT in both fine-tuning and training-from-scratch scenarios. We further find that minor architectural refinement during training from scratch can significantly improve extrapolation performance (see Section F). We present the comparison with LookHere in Section J. We report the result of 512×512 ImageNet in Section H.

**Arbitrary Aspect Ratio Extension.** Beyond generating square images, we evaluate the generalization abilities of LEDiT across different aspect ratios. Unlike FiT [28, 43], we do not apply multiple aspect ratio training techniques. Instead, we directly use the LEDiT-XL/2 model trained on the center-cropped 256×256 ImageNet dataset. This highlights the model's inherent generalization abilities. The quantitative results, reported in Table 3, demonstrate LEDiT's superiority. It achieves the best FID scores, with 20.29 and 18.82 at resolutions of 512×384 and 384×512, respectively, notably outperforming VisionNTK, VisionYaRN, and RIFLEx. These results confirm that LEDiT can generate high-quality images across diverse aspect ratios even without various aspect ratio training techniques. See Figure 20 for the qualitative comparison.

Table 3: Comparison of state-of-the-art extrapolation methods and our LEDiT trained on 256×256 ImageNet at arbitrary aspect ratios. We set CFG=1.5.

| Model | Resolution | FID↓ | sFID↓ | IS↑ | Prec.↑ | Rec.↑ |
|---|---|---|---|---|---|---|
| DiT-Sin/Cos PE | | 153.74 | 144.66 | 16.52 | 0.13 | 0.27 |
| DiT-VisionNTK | | 179.71 | 117.81 | 15.88 | 0.09 | 0.36 |
| DiT-VisionYaRN | 512×384 | 25.69 | 46.22 | 176.19 | 0.58 | 0.36 |
| DiT-RIFLEx | | 163.38 | 117.22 | 20.36 | 0.11 | 0.44 |
| FiTv2-VisionNTK | | 177.44 | 114.56 | 17.14 | 0.08 | 0.40 |
| FiTv2-VisionYaRN | | 56.04 | 51.05 | 81.96 | 0.35 | **0.47** |
| LEDiT | | **20.29** | **38.52** | **191.69** | **0.63** | 0.35 |
| DiT-Sin/Cos PE | | 158.21 | 139.80 | 16.98 | 0.14 | 0.26 |
| DiT-VisionNTK | | 150.70 | 110.67 | 25.88 | 0.14 | 0.41 |
| DiT-VisionYaRN | 384×512 | 22.02 | 48.72 | 202.03 | 0.61 | 0.35 |
| DiT-RIFLEx | | 143.84 | 116.93 | 27.42 | 0.14 | 0.45 |
| FiTv2-VisionNTK | | 177.44 | 114.56 | 17.14 | 0.08 | 0.40 |
| FiTv2-VisionYaRN | | 49.67 | 57.07 | 99.29 | 0.39 | **0.41** |
| LEDiT | | **18.82** | **42.64** | **205.38** | **0.64** | 0.36 |

VisionNTK, VisionYaRN, and RIFLEx. These results confirm that LEDiT can generate high-quality images across diverse aspect ratios even without various aspect ratio training techniques. See Figure 20 for the qualitative comparison.

**Text-to-image generation.** We further evaluate the performance of our method on the text-to-image generation task. Table 4 shows that LEMMDiT perform favorably compared to state-of-the-art extrapolation methods. Specifically, LEMMDiT achieves an FID of 29.89 and an sFID of 39.98, representing a substantial reduction compared to Vanilla MMDiT and RoPE-based variants. The CLIP Score [34] shows that LEDiT consistently outperforms other methods in terms of semantic coherence. In Figure 21, we present a qualitative comparison between LEMMDiT and other methods.

Vanilla MMDiT produces images with severe quality degradation. VisionNTK generates images with fine details but suffers from object duplication. VisionYaRN and RIFLEx yield more plausible object structures but lose fine-grained details. In contrast, LEDiT generates images with reasonable structures and rich details.

## 5 Conclusion

In this paper, we introduce a novel Diffusion Transformer, named Length-Extrapolatable Diffusion Transformer (LEDiT). LEDiT does not require explicit positional encodings such as RoPE. By combining causal attention and a locality enhancement module, LEDiT can implicitly encode positional information, which facilitates length extrapolation. Conditional and text-to-image gen-

Table 4: Text-to-image comparison of state-of-the-art extrapolation methods and our LEMMDiT trained on 256×256 COCO. The inference resolution is 512×512. We set CFG=2.0.

| Model | FID↓ | sFID↓ | IS↑ | Prec.↑ | Rec.↑ | CLIP↑ |
|---|---|---|---|---|---|---|
| MMDiT | 160.72 | 156.38 | 9.09 | 0.07 | 0.09 | 22.94 |
| MMDiT-ViNTK | 78.56 | 112.85 | 19.41 | 0.19 | 0.30 | 24.91 |
| MMDiT-ViYaRN | 163.88 | 93.39 | 10.35 | 0.11 | 0.30 | 23.37 |
| MMDiT-RIFLEx | 34.58 | 52.58 | 21.75 | 0.40 | **0.36** | 27.14 |
| LEMMDiT | **29.89** | **39.98** | **24.11** | **0.44** | 0.31 | **27.82** |

eration shows that LEDiT supports up to $4\times$ inference resolution scaling. Compared to previous extrapolation methods, we can generate images with more coherent object structures and richer details. We hope that LEDiT's principled departure from explicit positional encoding paradigms will not only advance the frontier of length extrapolation, but also inspire new perspectives on the foundational design space of transformer architecture.

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

# A Proof of Theorem 3.1

**Assumptions.** In Theorem 3.1, we demonstrate that Causal Attention introduces a position-dependent variance in attention outputs, allowing the Transformer to encode positional information implicitly.

To facilitate the subsequent derivations, we introduce the following assumptions:

**Stochastic Initialization Assumption:** We assume that the attention scores $S = \frac{QK^\top}{\sqrt{d_k}}$ are independently and identically distributed (i.i.d.). Analogously, the value $V$ is assumed to be i.i.d. with $E[V] = \mu_V$ and $\mathrm{Var}(V) = \sigma_V^2$.

**Mutual Independence:** We assume that the attention scores $\{S_{ij}\}$ and the value $V$ are mutually independent.

*Proof.* Consider a sequence of length $n$. For $1 \leq i, j \leq n$, the *causal attention matrix* $A_{ij}$ is defined by

$$A_{ij} = \begin{cases} \dfrac{\exp(S_{ij})}{\sum_{j'=1}^{i} \exp(S_{ij'})}, & i \geq j, \\ 0, & i < j. \end{cases} \tag{13}$$

Let $Z_{ij} = \exp(S_{ij})$ and $W_{ij} = \dfrac{Z_{ij}}{\sum_{j'=1}^{i} Z_{ij'}}$. Given the assumption on $S$, the elements $\{S_{ij}\}$ are assumed to be i.i.d., where

$$S_{ij} = \frac{1}{\sqrt{d_k}} \sum_{m=1}^{d_k} Q_{im} K_{jm}. \tag{14}$$

Then the attention output at position $i$ in dimension $l$ is $Y_{il} = \sum_{j=1}^{i} W_{ij} V_{jl}$. The variance of $(Y_{il})$ is :

$$\mathrm{Var}(Y_{il}) = \mathrm{Var}\Big(\sum_{j=1}^{i} W_{ij} V_{jl}\Big) = \sum_{j=1}^{i} \mathrm{Var}(W_{ij} V_{jl}), \tag{15}$$

since $W_{ij}$ and $V_{jl}$ are independent for each $i, j, l$. Furthermore, $\mathrm{Var}(W_{ij} V_{jl}) = E[W_{ij}^2] \sigma_V^2 + \mu_V^2 \mathrm{Var}(W_{ij})$. Hence,

$$\mathrm{Var}(Y_{il}) = \sum_{j=1}^{i} \big(E[W_{ij}^2] \sigma_V^2 + \mu_V^2 \mathrm{Var}(W_{ij})\big). \tag{16}$$

Because $Z_{ij}$ are i.i.d. and positive, we have $\sum_{j=1}^{i} W_{ij} = \dfrac{\sum_{j=1}^{i} Z_{ij}}{\sum_{j'=1}^{i} Z_{ij'}} = 1$, hence $E[W_{ij}] = 1/i$.

Besides, the normalized vector$(W_{i1}, W_{i2}, \ldots, W_{ii}) = \left(\dfrac{Z_{i1}}{\sum_{j'=1}^{i} Z_{ij'}}, \ldots, \dfrac{Z_{ii}}{\sum_{j'=1}^{i} Z_{ij'}}\right)$ can be approximated by a $\mathrm{Dirichlet}(1, \ldots, 1)$ distribution. This approximation is conceptually aligned with the analytic framework proposed by Hobbhahn [19], which establishes a mapping from a distribution over logits to a Dirichlet distribution on the corresponding softmax outputs. Reasonably, whenever the exponentials $\{\exp(S_{ij})\}$ do not differ too sharply and remain roughly exchangeable, this leads to the uniform-symmetric Dirichlet scenario. In practice, it provides a convenient closed-form $E[W_{ij}^2] = \frac{2}{i(i+1)}$.

Then $\mathrm{Var}(W_{ij}) = E[W_{ij}^2] - (E[W_{ij}])^2 = \frac{2}{i(i+1)} - \frac{1}{i^2}$. Substituting into the sum, one obtains

$$\mathrm{Var}(Y_{il}) = \sum_{j=1}^{i} \Big(\frac{2}{i(i+1)} \sigma_V^2 + \mu_V^2 \Big[\frac{2}{i(i+1)} - \frac{1}{i^2}\Big]\Big) = \frac{2}{i+1} \sigma_V^2 + \frac{i-1}{i(i+1)} \mu_V^2. \tag{17}$$

As $i$ increases, we can approximate $\frac{i-1}{i(i+1)} \approx \frac{1}{i+1}$, leading to the reasonable approximation

$$\mathrm{Var}(Y_{il}) \approx \frac{C}{i+1}, \tag{18}$$

where the constant $C = 2\sigma_V^2 + \mu_V^2$.

$\square$

## B   Justification of Mutual Independence Assumption

We observe that the correlation between the attention matrix and the value vectors is low during the early stages of denoising. This empirical observation supports the validity of independence assumption. As discussed in the main paper, our analysis primarily focuses on the early stage, where we show that causal attention mainly encodes positional information.

The input sequence $x = [x_1, ..., x_n]$ can be approximated as i.i.d. Gaussian noise in the early denoising stage. Each $x_i$ is an independent Gaussian vector. The queries and keys are computed as: $q_i = x_i W_q, k_j = x_j W_k$ Since the $x_i$ are i.i.d., the sets $q_i$ are also i.i.d. after linear transformation, the same as $k_j$. The attention score is: $S_{ij} = q_i^T k_j$. Because the $k_j$ are i.i.d., the set $S_{ij}$ for $j = 1, ... n$ are identically distributed random variables. After applying softmax $A = \text{Softmax}(S)$, the attention matrix $A_{ij}$ approaches a uniform distribution, and $\mathbb{E}[A_{ij}] = 1/n$. From this perspective, the attention matrix and the value have low correlation: changes in the value vectors do not significantly affect the attention matrix, resulting in low correlation. We conduct a toy experiment with a sequence length of 256 and a head hidden dimension of 72, consistent with the DiT-XL/2 configuration. We randomly initialized $W_q, W_k, W_v$, generated random Gaussian noise $x$, and computed the correlation coefficient between the attention scores $S$ and the values $V$ over 1000 trials. The average correlation coefficient was 0.03, indicating very low correlation. This provides theoretical support for our assumption.

In practice, we also observe several layers show low correlation between the attention matrix and the value vectors during the early denoising stages of a trained diffusion transformer. As shown in Figure 10, we report the correlation coefficients between the attention matrix and the value vectors across different timesteps and layers. The correlation remains low in the early stages of denoising. While the correlation gradually increases in later stages—where the independence assumption no longer holds—we have demonstrated in the main paper that causal attention primarily encodes positional information during the early denoising steps. Therefore, this does not affect the validity of our justification.

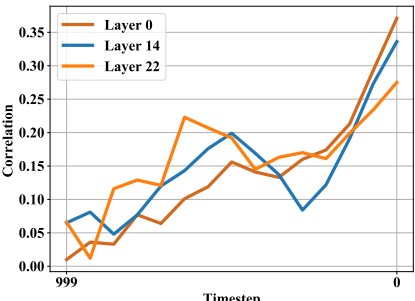

Figure 10: Correlation across timestep.

## C   Variance Distribution Across Timesteps

As shown in Figure 11, we present the variance distribution of causal attention outputs across different timesteps and layers. During the early stages of denoising, the variance distribution mainly follows our proposed theorem. In the late denoising stage, especially $T < 100$, the variance distribution deviates from the theorem and becomes irregular. We attribute this to the higher independence among values $V$ in the early stages, which aligns with the theorem's assumptions. In the later stages, the increasing correlation between tokens (position and semantic relationships) violates the assumptions. This raises a question: What is the potential impact of variance deviation in later denoising steps on image quality? To further investigate this, we conduct additional experiments. Specifically, we introduce a switching threshold $T'$ in the denoising process. Given that the denoising timestep $T$ decreases from 1000 to 0, we design the attention mechanism as follows: when $T \geq T'$, we use

causal attention; when $T < T'$, we switch to self-attention. In this setup, a smaller $T'$ corresponds to switching later in the denoising process.

We choose different values $T'$ to train an LEDiT-XL/2 on 256×256 ImageNet and use it to generate 512×512 images. As shown in Figure 12, we find a clear trend: the later the switch, namely the smaller the $T'$, the better the generated image quality. Notably, when $T' = 100$, the generated images are comparable to those of LEDiT with full causal attention ($T' = 0$). This suggests that the positional information is primarily acquired in the early phase, and the causal attention with variance deviation in the later steps ($T < 100$) has minor effects on the resolution extrapolation ability.

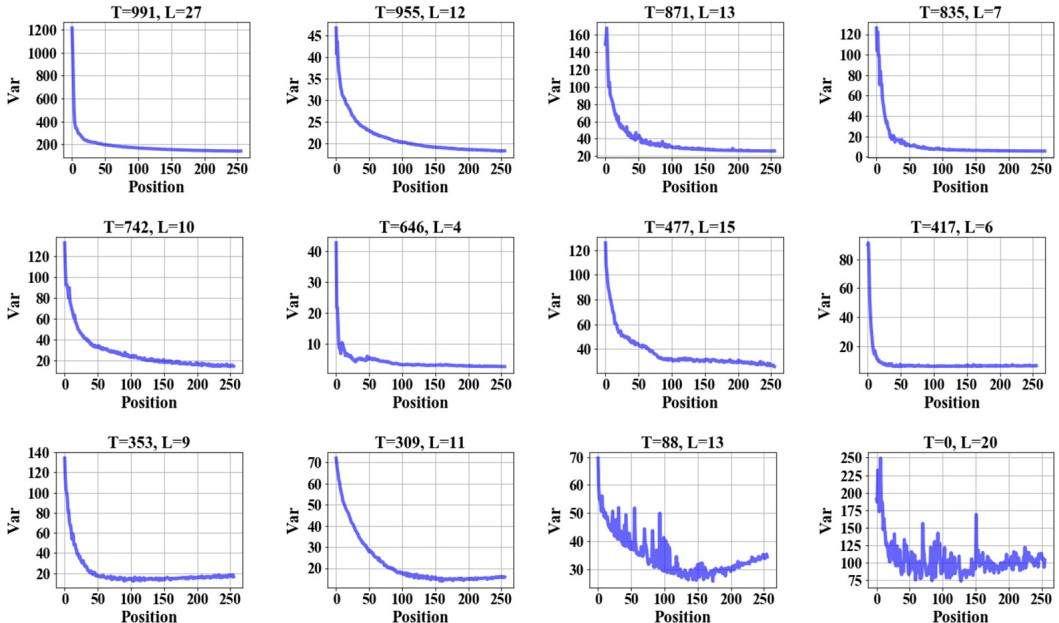

Figure 11: Variance distribution across different timesteps and layers.

$T' = 400$      $T' = 200$      $T' = 100$      $T' = 0$

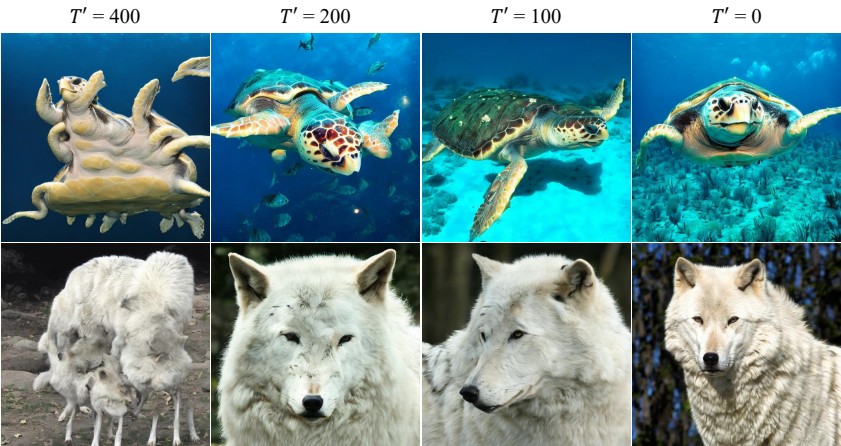

Figure 12: Switching threshold ablations. The resolution is 512×512. We set CFG=4.0. Variance deviation in the later steps ($T < 100$) has minor effects on the resolution extrapolation ability.

## D    Variance Distribution of Variant d

The variance distribution under the "Mask Lower-right Corner" order is consistent with our main findings. As this is a 2D scan variant, the variance is expected to decrease progressively along the

Table 5: Positional index regression.

| Method | 1D Position Regression | | 2D Position Regression | |
|---|---|---|---|---|
| | Training Loss | Test Error | Training Loss | Test Error |
| DiT-NoPE | 5091.24 | 5265.85 | 12.57 | 13.30 |
| **DiT-Cau. Atten.** | **97.42** | **112.08** | **1.20** | **1.36** |

height or width axis. Figure 13 demonstrates that the 2D causal scan variants still exhibit the existence of causal attention in DiT that satisfies the conditions outlined in our theorem.

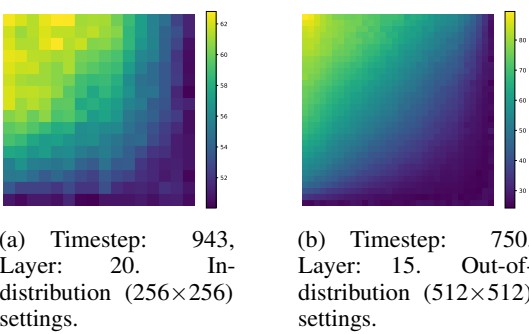

(a) Timestep: 943, Layer: 20. In-distribution (256×256) settings.

(b) Timestep: 750, Layer: 15. Out-of-distribution (512×512) settings.

Figure 13: Variance distribution of variant d.

# E   Positional Index Regression

We further conduct positional index regression experiment to verify that causal attention can encode positional information. Specifically, we trained an MLP to predict the position index of each token using the outputs of causal attention from a well-trained DiT as input. We conducted experiments on both 1D and 2D position regression tasks on ImageNet-256×256 to validate the effectiveness.

For 1D causal attention, the MLP predicts the 1D position index of each token (e.g., 1, 2, ... 256). For 2D causal attention, the MLP predicts the 2D position index (e.g., 1,1, 1,2,...,16,16) for each token. The MLP is trained using L2 loss.

We performed these experiments using DiT with 1D causal attention (variant (a) in the main paper) and DiT with 2D causal attention (variant (d)), and compared the results with DiT-NoPE, which cannot encode positional information. If the causal attention variants outperform DiT-NoPE, it demonstrates that the outputs of causal attention contain implicit positional information.

During inference, we generated images using both DiT with causal attention and DiT-NoPE. At each of the 250 denoising steps, the MLP predicts the positional index from the features output by causal attention. We generated 100 images, resulting in 2,500 tests in total. We report the L2 loss between the predicted and ground truth position indices. Since the MLP is trained to predict positional indices, it cannot generalize to unseen positional indices when extrapolating to higher resolutions. Therefore, we perform inference with the MLP at the training resolution. Nevertheless, the significant performance gap compared to NoPE provides strong evidence that causal attention can implicitly encode positional information. As shown in the table below, causal attention demonstrates a significant advantage over NoPE in both training loss and test error. This indicates that the position regressor can effectively learn positional information from the outputs of causal attention, providing further evidence that causal attention can implicitly encode positional information.

# F   Architecture Refinement

We observe that minor architectural modifications to LEDiT during training from scratch can significantly improve extrapolation performance, as detailed in Table 6. (i) Adding a layer normalization after convolution stabilizes training and enhances extrapolation. (ii) Using only a single causal layer achieves strong extrapolation, slightly outperforming the 14-layer setting. (iii) The multi-dilation

strategy reduces sFID but leads to a slight increase in FID. This finding contrasts with the fine-tuning scenario in Table 1c. We hypothesize that, during training from scratch, the dilation perturbations may hinder convergence, whereas in the fine-tuning setting, where the model is already well-trained, dilation has less impact on convergence. A promising future direction is to adopt a progressive multi-dilation strategy, which we leave for future work.

Table 6: Ablation study on architecture refinement. The inference resoluton is $512\times512$. The models are trained on $256\times256$ ImageNet, and we report results with 10K samples.

| Causal Layers | Conv Post Norm | Multi-dilation | FID↓ | sFID↓ | IS↑ | Prec.↑ | Rec.↑ |
|---|---|---|---|---|---|---|---|
| 14 | | ✓ | 59.32 | 78.35 | 62.13 | 0.42 | 0.36 |
| 14 | ✓ | ✓ | 40.45 | 72.98 | 104.24 | 0.56 | 0.39 |
| 14 | ✓ | | 39.84 | 80.41 | 104.26 | 0.55 | 0.38 |
| 1 | ✓ | | 36.79 | 76.98 | 112.82 | 0.56 | 0.39 |

# G   In-distribution Comparison

In the main paper, we compared the results of various methods at resolutions higher than the training resolution. In this section, we compare the performance of LEDiT-XL/2 and LEMMDiT at the training resolution. Tables 7 and 8 shows the result of LEDiT-XL/2 on ImageNet and LEMMDiT on COCO. The FID of LEDiT and LEMMDiT increases slightly at $256\times256$. Although both LEDiT (with causal attention) and DiT (with standard self-attention) have approximately the same number of parameters, their computational complexities differ substantially. For an input sequence of length $L$ and hidden dimension $d$, the self-attention mechanism computes attention scores for all possible pairs, resulting in a per-layer computational complexity of $\mathcal{O}(L^2 d)$. In contrast, causal attention restricts each position to attend only to previous positions (including itself), leading to a reduced number of attention computations. Specifically, the total number of attention weights is reduced from $L^2$ to $L(L+1)/2$, and the corresponding computational complexity becomes $\mathcal{O}(L^2 d/2)$. This halves the theoretical compute cost compared to self-attention, i.e., $\frac{\mathcal{O}(L^2 d)}{\mathcal{O}(L^2 d/2)} = 2$. While this reduction improves efficiency theoretically, it may also limit the model's ability to capture long-range dependencies, which can explain the slight performance gap between LEDiT and DiT. Nevertheless, as illustrated in Figure 14, LEDiT still produces high-fidelity samples, demonstrating that causal attention still achieves competitive generative quality despite its lower computational complexity.

Table 7: Comparison of performance on $256\times256$ resolution. The models are trained on $256\times256$ ImageNet. We set CFG=1.5. We report results with 50K samples.

| Model | Resolution | FID↓ | sFID↓ | IS↑ | Prec.↑ | Rec.↑ |
|---|---|---|---|---|---|---|
| DiT-Sin/Cos PE | | 2.27 | 4.60 | 278.24 | 0.83 | 0.57 |
| DiT-RoPE | | 2.33 | 4.58 | 272.02 | 0.83 | 0.58 |
| DiT-Learnable PE | $256\times256$ | 2.38 | 4.69 | 275.05 | 0.82 | 0.58 |
| DiT-LH-180 | | 2.54 | 4.94 | 248.47 | 0.82 | 0.57 |
| LEDiT | | 2.38 | 4.58 | 268.66 | 0.83 | 0.58 |

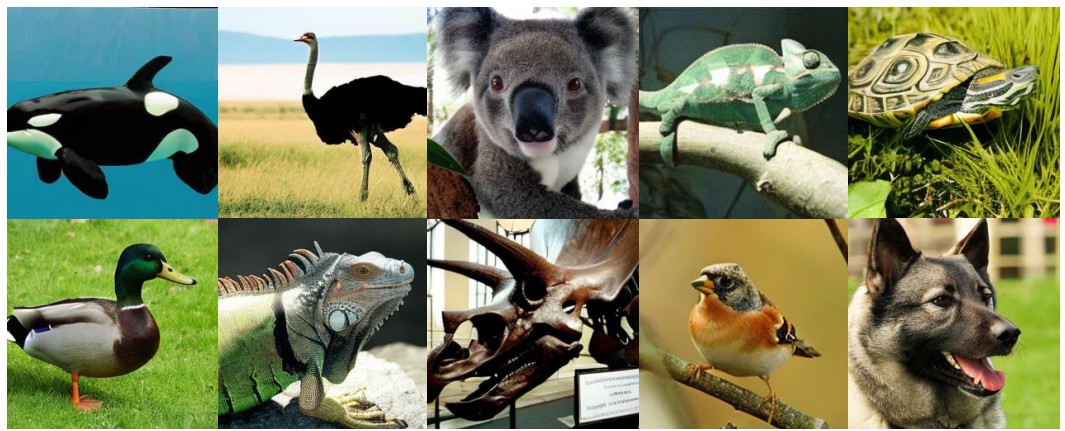

Figure 14: 256×256 samples generated from our LEDiT-XL/2 trained on ImageNet 256×256 resolution with CFG = 4.0.

Table 8: Comparison of performance on 256×256 resolution. The models are trained on 256×256 COCO. We set CFG=2. We report results with 40,504 samples.

| Model | Resolution | FID↓ | sFID↓ | IS↑ | Prec.↑ | Rec.↑ |
|---|---|---|---|---|---|---|
| MMDiT | | 6.32 | 11.77 | 30.01 | 0.65 | 0.49 |
| MMDiT-RoPE | 256×256 | 5.39 | 11.68 | 32.32 | 0.67 | 0.50 |
| LEMMDiT | | 6.35 | 11.61 | 31.54 | 0.65 | 0.48 |

## H  512×512 ImageNet

We fine-tune a new LEDiT-XL/2 model on 512×512 ImageNet for 100K iterations using the same hyperparameters as the 256×256 model. The qualitative comparison among vanilla DiT, VisionNTK, VisionYaRN, and LEDiT is shown in Figure 9. As resolution increases from 512×512 to 1024×1024, vanilla DiT exhibits further quality degradation, with significant noise artifacts. FiTv2-VisionNTK generates images with duplicated objects, while FiTv2-VisionYaRN produces blurry images with severe high-frequency detail loss. DiT-VisionNTK, VisionYaRN, RIFLEx generate higher-quality images but exhibit object duplication in local structures. In contrast, LEDiT maintains more realistic structures and finer details. The quantitative results are reported in Table 9. Due to the heavy quadratic computational burden, we generate 10K images for evaluation. LEDiT consistently achieves superior metric scores across all resolution settings. For instance, at a resolution of 768×768, LEDiT improves the previous best FID of 28.94 (achieved by DiT-VisionNTK) to 21.75.

Table 9: Comparison of state-of-the-art extrapolation methods and our LEDiT trained on 512×512 ImageNet at various resolutions beyond the training image size. We set CFG=1.5.

| Model | 768×768 | | | | | 896×896 | | | | | 1024×1024 | | | | |
|---|---|---|---|---|---|---|---|---|---|---|---|---|---|---|---|
| | FID↓ | sFID↓ | IS↑ | Prec.↑ | Rec.↑ | FID↓ | sFID↓ | IS↑ | Prec.↑ | Rec.↑ | FID↓ | sFID↓ | IS↑ | Prec.↑ | Rec.↑ |
| DiT-Sin/Cos PE | 159.52 | 187.92 | 7.76 | 0.12 | 0.24 | 229.93 | 217.70 | 3.27 | 0.03 | 0.08 | 281.57 | 240.17 | 2.16 | 0.01 | 0.03 |
| DiT-VisionNTK | 28.94 | 96.37 | 142.35 | 0.67 | 0.53 | 64.41 | 139.97 | 64.52 | 0.48 | 0.47 | 109.31 | 170.58 | 25.31 | 0.29 | 0.39 |
| DiT-VisionYaRN | 29.46 | 61.37 | 161.32 | 0.66 | 0.53 | 65.58 | 91.48 | 83.21 | 0.50 | **0.51** | 104.62 | 118.03 | 43.04 | 0.35 | **0.47** |
| DiT-RIFLEx | 31.10 | 80.71 | 134.01 | 0.66 | 0.49 | 64.84 | 115.61 | 62.90 | 0.48 | 0.47 | 114.84 | 152.90 | 23.87 | 0.27 | 0.36 |
| FiTv2-VisionNTK | 251.73 | 195.83 | 3.44 | 0.02 | 0.12 | 309.13 | 230.84 | 2.54 | 0.01 | 0.01 | 349.76 | 240.17 | 2.43 | 0.01 | 0.01 |
| FiTv2-VisionYaRN | 51.13 | 64.48 | 70.40 | 0.49 | **0.62** | 215.72 | 175.75 | 6.33 | 0.06 | 0.41 | 327.26 | 217.01 | 2.91 | 0.01 | 0.08 |
| LEDiT | **21.75** | **49.81** | 176.26 | **0.71** | 0.52 | **48.64** | **73.25** | 97.54 | **0.56** | 0.50 | **91.11** | **108.70** | 48.13 | **0.40** | 0.44 |

## I  Comparison with Learnable Positional Embeddings

Learnable positional embeddings have been widely adopted in the original ViT [9] and Swin Transformer [27]. We replace the Sin/Cos PE in DiT with learnable positional embeddings to conduct a comparison with our method. For length extrapolation, we interpolate the learnable positional

embeddings to higher resolutions to ensure compatibility. We report the in-distribution performance in Table 7 and the out-of-distribution performance in Table 10. At the training resolution, Learnable PE and LEDiT exhibit nearly comparable performance. When extrapolating to 512×512, we observe a significant drop for Learnable PE. This is likely because the interpolated positional embeddings at new spatial locations are not seen during training, leading to degradation.

Table 10: Comparison with learnable positional embeddings on 512×512 resolution. The models are trained on 256×256 ImageNet, and we report results with 10K samples.

| Model | FID↓ | sFID↓ | IS↑ | Prec.↑ | Rec.↑ |
|---|---|---|---|---|---|
| DiT-Learnable PE | 208.45 | 139.38 | 5.23 | 0.02 | 0.02 |
| LEDiT | **35.86** | **67.97** | **139.91** | **0.52** | **0.51** |

## J    Comparison with LookHere

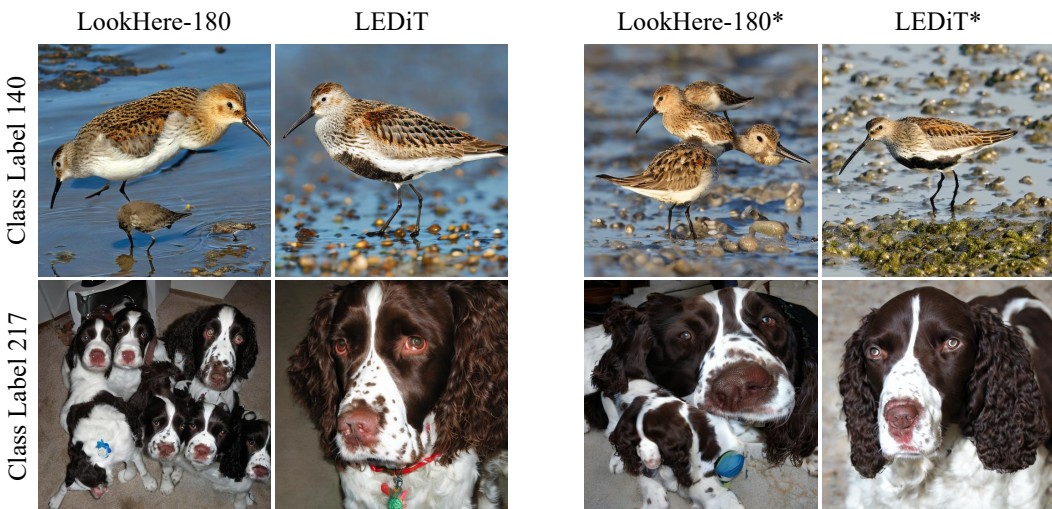

Figure 15: Comparison with LookHere at 512×512 resolution. The models are trained on 256×256 ImageNet. We set CFG=4.0. * indicates training from scratch. Best viewed when zoomed in.

Both LEDiT and LookHere [12] explore causal attention to enhance length extrapolation. LookHere carefully designs various combinations of causal masks to provide directional inductive biases. It introduces AliBi [33] to penalize attention scores and demonstrates extrapolation improvement. LookHere conduct extensive experiments to demonstrate the extrapolation capabilities in classification.

However, we clarify that there are key differences. We provide a rigorous theoretical framework that explains why causal attention is capable of encoding positional information and enabling length extrapolation. This not only facilitates effective image generation with robust extrapolation capabilities, but also offers valuable insights into the underlying mechanisms of length extrapolation. In contrast, LookHere does not investigate the reasons behind causal attention's ability to encode positional information; it merely states that attention with 2D masks can "limit the distribution shift that attention heads face when extrapolating".

We directly adapt LookHere to image generation tasks and find it does not yield effective length extrapolation. We select LookHere-180—the best-performing variant—as the representative method and compare its performance with LEDiT. We (i) train the randomly initialized LookHere/LEDiT for 400K steps or (ii) fine-tune LookHere/LEDiT for 100K steps. Quantitative results are presented in Table 11. In the fine-tuning scenario, LEDiT outperforms LookHere. In the training-from-scratch scenario, LookHere achieves a lower sFID, while LEDiT achieves a lower FID. Specifically, sFID leverages intermediate spatial features from the Inception network, capturing fine-grained image details, whereas FID is computed using the spatially-pooled layer, reflecting more global structures. We interpret that the lower sFID of LookHere indicates fine-grained image details, while the lower FID of LEDiT suggests more coherent object structures. As shown in Figure 15, when extrapolated

to 512×512 resolution, samples generated by LookHere preserve fine image details but suffer from severe object duplication, whereas those generated by LEDiT exhibit more coherent structures, indicating that LEDiT shows better extrapolation performance. We believe a promising direction is to integrate LookHere and LEDiT, aiming to generate images with both high-quality details and coherent object structures.

Table 11: Comparison with LookHere on 512×512 resolution. The models are trained on 256×256 ImageNet. We set CFG=1.5. We report results with 10K samples. * indicates training from scratch.

| Model | FID↓ | sFID↓ | IS↑ | Prec.↑ | Rec.↑ |
|---|---|---|---|---|---|
| DiT-LH-180 | 66.93 | 83.09 | 82.39 | 0.39 | 0.36 |
| LEDiT | **35.86** | **67.97** | **139.91** | **0.52** | **0.51** |
| DiT-LH-180* | 41.97 | **71.42** | 111.44 | 0.56 | 0.33 |
| LEDiT* | **36.79** | 76.98 | **112.82** | 0.56 | **0.39** |

## K 16× Length Extrapolation

We evaluate LEDiT trained on ImageNet-256×256 and extrapolated to 1024×1024 resolution (a 16× length extrapolation). As shown in Table 12, LEDiT outperforms other methods. However, the high FID suggests that aggressive resolution extrapolation remains challenging and warrants further exploration.

Table 12: Comparison with other method on 1024×1024 resolution. The models are trained on 256×256 ImageNet, and we report results with 10K samples.

| Model | FID↓ | sFID↓ | IS↑ | Prec.↑ | Rec.↑ |
|---|---|---|---|---|---|
| DiT-Sin/Cos PE | 281.57 | 240.17 | 2.16 | 0.01 | 0.02 |
| DiT-Learnable PE | 284.07 | 230.15 | 2.41 | 0.08 | 0.01 |
| DiT-VisionNTK | 333.01 | 244.23 | 1.96 | 0.22 | 0.00 |
| DiT-VisionYaRN | 228.41 | 199.62 | 7.60 | 0.03 | 0.09 |
| DiT-RIFLEx | 335.30 | 214.46 | 4.86 | 0.01 | 0.12 |
| FiTv2-VisionNTK | 342.54 | 260.28 | 2.75 | 0.01 | 0.00 |
| FiTv2-VisionYaRN | 338.12 | 241.12 | 2.93 | 0.01 | 0.00 |
| LEDiT | **212.97** | **169.88** | **10.14** | **0.05** | **0.14** |

## L FID over Fine-tuning Steps

We plot FID over fine-tuning steps from 25K to 200K at both training resolution (256×256) and beyond (512×512), as shown in Figure 16. At 256x256 resolution, FID generally decreases with more fine-tuning steps. At 512x512, performance plateaus around 50K steps, then gradually increases with further fine-tuning. We chose 100K steps to balance performance at in-distribution and out-of-distribution.

## M Ablation Study on Attention Scaling

Following the widely adopted practice in length extrapolation, we also apply attention scaling [22]. Figure 17 shows the ablation of attention scaling. Without attention scaling, LEDiT can still generate reasonable images. Attention scaling primarily improves image quality and mitigates local structural issues (e.g., the dog's mouth).

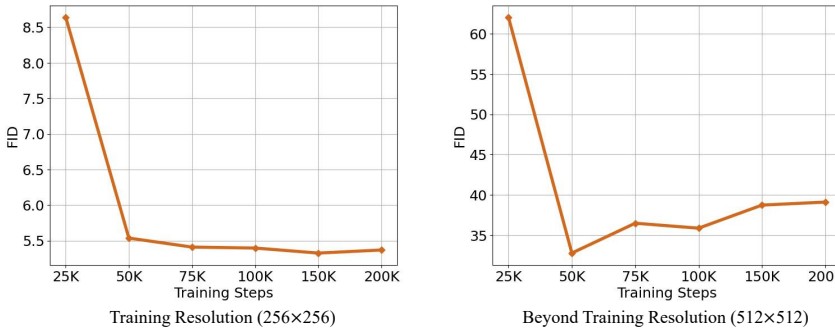

Figure 16: FID-10K over fine-tuning steps of LEDiT at the training resolution (256×256) and beyond the training resolution (512×512).

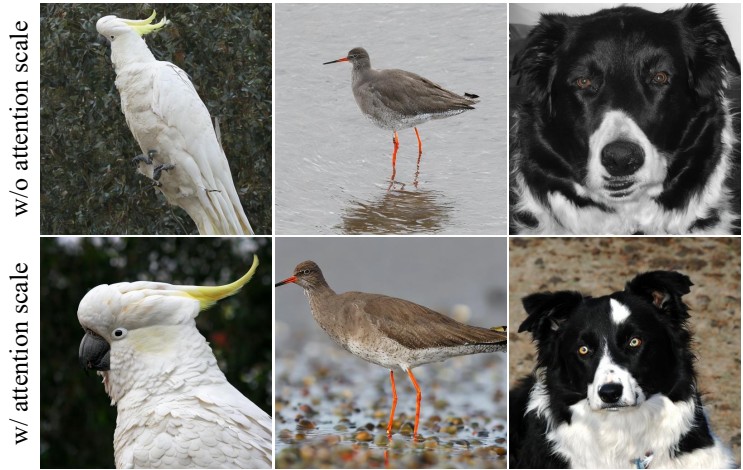

Figure 17: Attention scaling ablation. The resolution is 512×512 generated by LEDiT-XL/2 trained on ImageNet-256×256. We set CFG=4.0.

## N   Ablation Study on CFG

Following prior work, FiT, we set CFG=1.5. Additionally, we test model performance across various CFGs, where CFG=1 indicates no classifier-free guidance. As shown in Table 13, LEDiT outperforms VisionNTK, VisionYaRN and RIFLEx across different CFGs, demonstrating LEDiT's robust extrapolation capability.

Table 13: Ablation study on CFGs. The models are trained on 256×256 ImageNet. The inference resolution is 512×512 and we report results with 10K samples.

| Model | CFG = 1.0 | | | | | CFG = 1.5 | | | | | CFG = 2.0 | | | | |
|---|---|---|---|---|---|---|---|---|---|---|---|---|---|---|---|
| | FID↓ | sFID↓ | IS↑ | Prec.↑ | Rec.↑ | FID↓ | sFID↓ | IS↑ | Prec.↑ | Rec.↑ | FID↓ | sFID↓ | IS↑ | Prec.↑ | Rec.↑ |
| DiT-VisionNTK | 279.14 | 173.27 | 6.17 | 0.02 | 0.16 | 251.85 | 153.40 | 8.73 | 0.04 | 0.26 | 213.99 | 134.39 | 11.89 | 0.06 | 0.27 |
| DiT-VisionYaRN | 125.73 | 123.19 | 32.78 | 0.19 | 0.48 | 53.75 | 76.62 | 107.75 | 0.41 | **0.53** | 27.12 | 54.04 | 211.85 | 0.61 | 0.46 |
| DiT-RIFLEx | 302.35 | 197.81 | 7.98 | 0.03 | 0.11 | 256.1 | 172.54 | 10.54 | 0.04 | 0.18 | 214.30 | 149.61 | 13.48 | 0.05 | 0.21 |
| FiTv2-VisionNTK | 301.94 | 214.17 | 3.55 | 0.00 | 0.08 | 265.40 | 198.17 | 6.58 | 0.01 | 0.03 | 205.64 | 150.89 | 9.22 | 0.03 | 0.23 |
| FiTv2-VisionYaRN | 246.06 | 180.58 | 8.43 | 0.04 | 0.29 | 163.27 | 139.45 | 20.42 | 0.12 | 0.35 | 97.49 | 95.19 | 46.50 | 0.22 | **0.52** |
| LEDiT | **86.90** | **107.87** | **45.19** | **0.26** | **0.54** | **35.86** | **67.97** | **139.91** | **0.52** | 0.51 | **20.51** | **45.71** | **250.96** | **0.69** | 0.44 |

## O   More Training Steps

In this section, we use LEDiT without architecture refinement. We further extended the training from scratch to 600K and 1000K steps, as presented in Table 14. LEDiT consistently outperforms VisionNTK, VisionYaRN and RIFLEx, and we do not observe FID saturation.

Table 14: Comparison of state-of-the-art extrapolation methods when training from scratch on 256×256 ImageNet. We extend the training steps to 600K and 1000K. The inference resolution is 512×512. We report results with 10K samples.

| Model | Training Steps (K) | FID↓ | sFID↓ | IS↑ | Prec.↑ | Rec.↑ |
|---|---|---|---|---|---|---|
| DiT-Sin/Cos PE | 600 | 244.22 | 193.22 | 2.74 | 0.19 | 0.04 |
| DiT-VisionNTK | 600 | 183.16 | 146.21 | 16.18 | 0.09 | 0.33 |
| DiT-VisionYaRN | 600 | 118.44 | 123.36 | 36.68 | 0.20 | **0.43** |
| DiT-RIFLEx | 600 | 132.27 | 129.26 | 25.94 | 0.15 | 0.40 |
| LEDiT | 600 | **56.34** | **72.14** | **74.10** | **0.44** | 0.38 |
| DiT-VisionNTK | 1000 | 190.65 | 151.52 | 14.19 | 0.09 | 0.35 |
| DiT-VisionYaRN | 1000 | 131.03 | 127.05 | 32.13 | 0.17 | **0.44** |
| DiT-RIFLEx | 1000 | 160.42 | 117.45 | 21.37 | 0.11 | 0.40 |
| LEDiT | 1000 | **54.29** | **74.53** | **79.35** | **0.45** | 0.42 |

## P   Performance on Smaller Models

We conduct experiments on smaller models, namely DiT-B and DiT-S, under the same training settings as LEDiT-XL/2. As shown in Table 15, LEDiT consistently outperforms state-of-the-art extrapolation methods across all evaluation metrics in DiT-B, and shows substantial improvements in key metrics in DiT-S, with comparable FID to DiT-RoPE-NTK. It delivers stable performance gains across DiT-S, DiT-B, and DiT-XL, demonstrating robustness and scalability. These results highlight LEDiT's strong length extrapolation capabilities and generalizability to different model scales.

Table 15: Comparison of performance on 512×512 resolution using DiT-B and DiT-S. The models are trained on 256×256 ImageNet for 400K steps, and we report results with 10K samples.

| Model | FID↓ | sFID↓ | IS↑ | Prec.↑ | Rec.↑ |
|---|---|---|---|---|---|
| DiT-S-Sin/Cos PE | 253.20 | 186.79 | 2.36 | 0.02 | 0.01 |
| DiT-S-VisionNTK | **121.42** | 197.59 | 11.61 | 0.12 | **0.23** |
| DiT-S-VisionYaRN | 161.41 | 129.67 | 13.40 | 0.11 | 0.20 |
| DiT-S-RIFLEx | 313.10 | 192.36 | 6.34 | 0.02 | 0.20 |
| LEDiT-S | 124.71 | **97.76** | **18.57** | **0.16** | 0.22 |
| DiT-B-Sin/Cos PE | 214.05 | 188.24 | 3.27 | 0.02 | 0.07 |
| DiT-B-VisionNTK | 164.51 | 193.89 | 6.98 | 0.09 | 0.24 |
| DiT-B-VisionYaRN | 126.52 | 132.84 | 21.37 | 0.18 | **0.31** |
| DiT-B-RIFLEx | 433.49 | 225.40 | 4.00 | 0.01 | 0.09 |
| LEDiT-B | **86.63** | **84.93** | **35.21** | **0.29** | 0.30 |

## Q   More Ablation Visualization

We present all visualizations of the ablation study in Figure 18.

## R   More Qualitative Visualization

We present more comparison results with DiT-Sin/Cos PE, DiT-VisionNTK, DiT-VisionYaRN, DiT-REFLEx, FiTv2-VisionNTK, and FiTv2-VisionYaRN to demonstrate the effectiveness of LEDiT, as shown in Figure 19, Figure 20, and Figure 21. LEDiT outperforms other methods in both fidelity and local details.

## S   Additional Samples

We present more samples generated by LEDiT-XL/2 in Figure 22 and Figure 23.

# T  Limitations and Future Work

Constrained by limited resources, we train LEDiT only on the ImageNet and COCO datasets. We did not test how LEDiT will perform on a larger-scale dataset such as LAION-5B [38]. The generative capabilities of LEDiT when training with higher resolutions have not been explored. Subsequent research can focus on how to integrate LEDiT into modern powerful diffusion models or LLMs to achieve more amazing outcomes. Moreover, other learnable modules capable of capturing positional information, similar to causal attention, should be explored to further enhance the performance of length extrapolation.

# U  Broader Impacts

Our diffusion-based approach can advance generative modeling, enabling applications in image synthesis, data augmentation, and scientific discovery, which may benefit research and industry. At the same time, our method could be misused for generating misleading or harmful content, such as deepfakes or synthetic data for malicious purposes. We discuss these risks and suggest possible mitigation strategies.

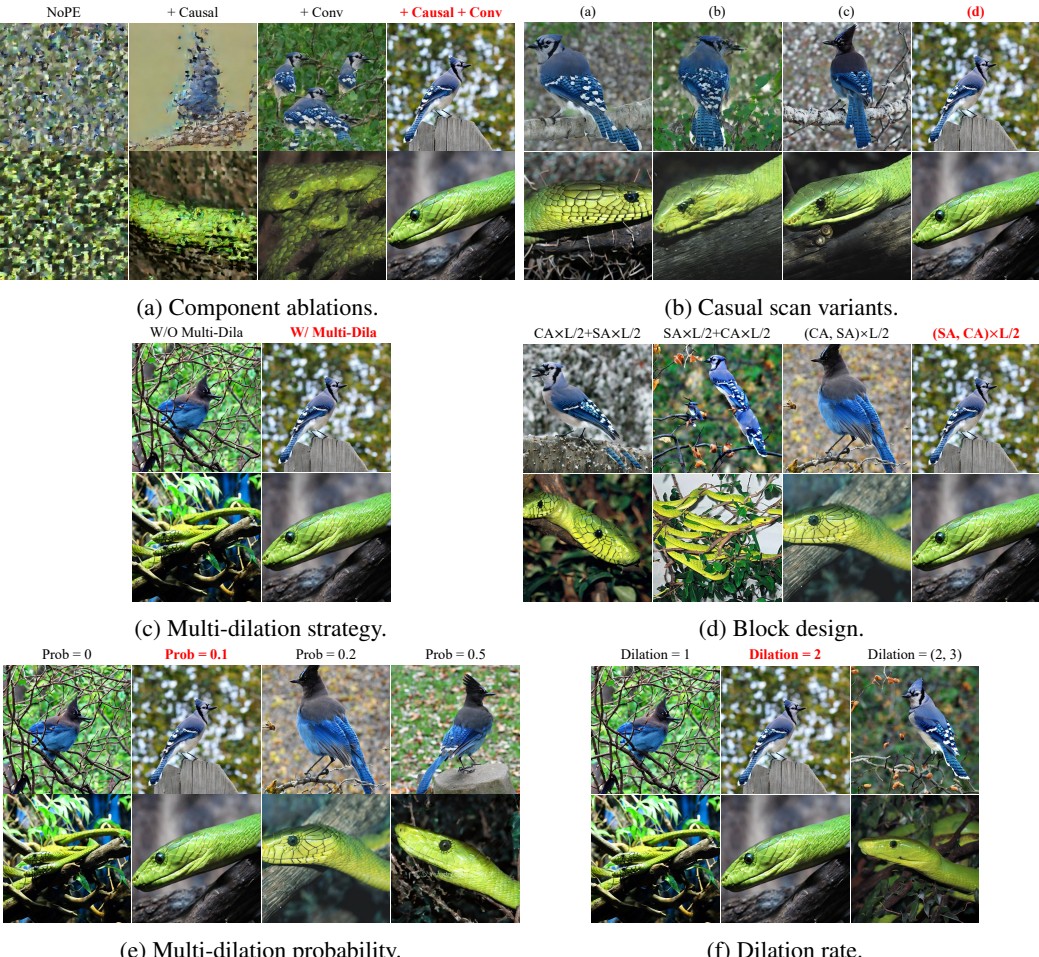

Figure 18: Ablation visualization of Table 1. We set CFG=4.0. Best viewed when zoomed in.

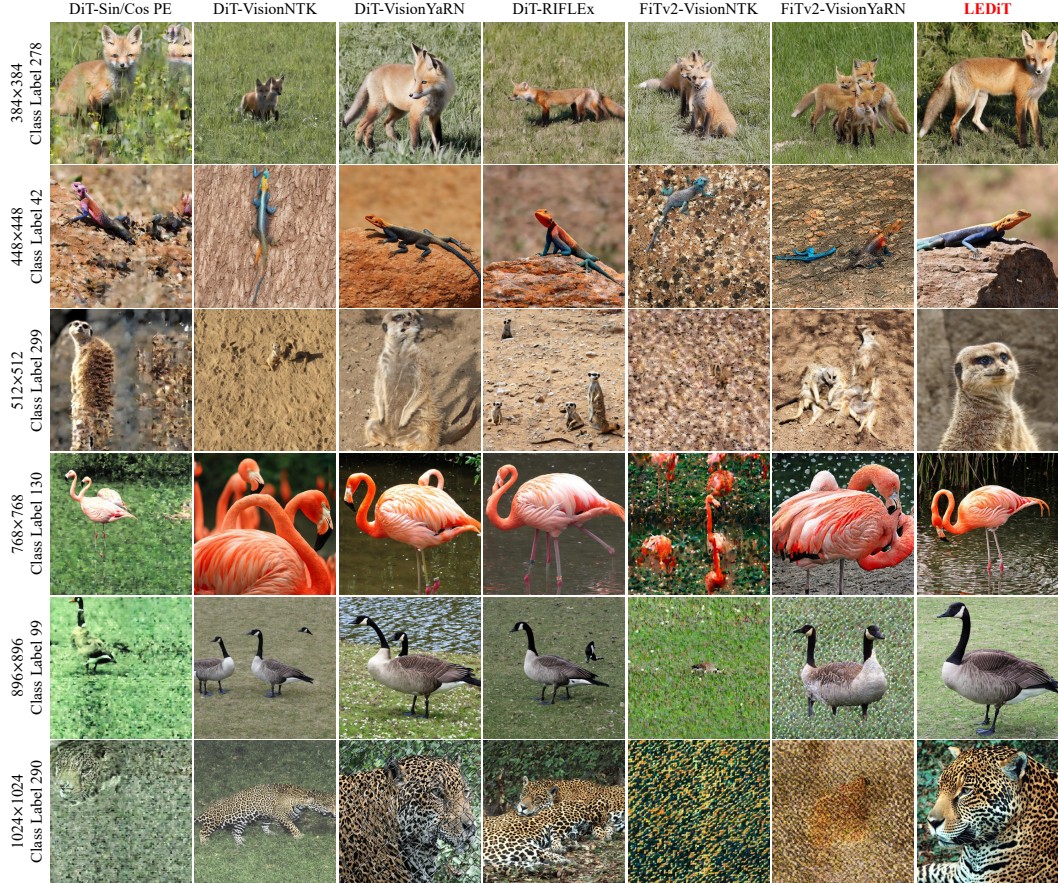

Figure 19: More qualitative comparison with other methods. The resolution and class label are located to the left of the image. We use the model trained on 256×256 ImageNet to generate images with resolutions less than or equal to 512×512, and the model trained on 512×512 ImageNet to generate images with resolutions greater than 512×512 and less than or equal to 1024×1024. We set CFG=4.0. Best viewed when zoomed in.

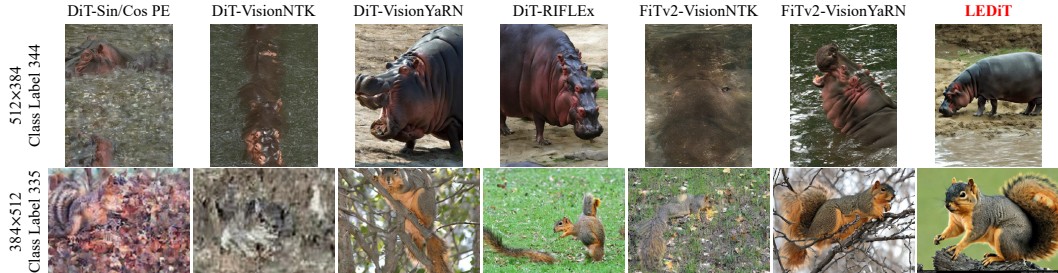

Figure 20: More qualitative comparison with other methods on generating non-square images. The resolution and class label are located to the left of the image. We use the model trained on 256×256 ImageNet to generate images at 512×384 and 384×512 resolutions. We set CFG=4.0. Best viewed when zoomed in.

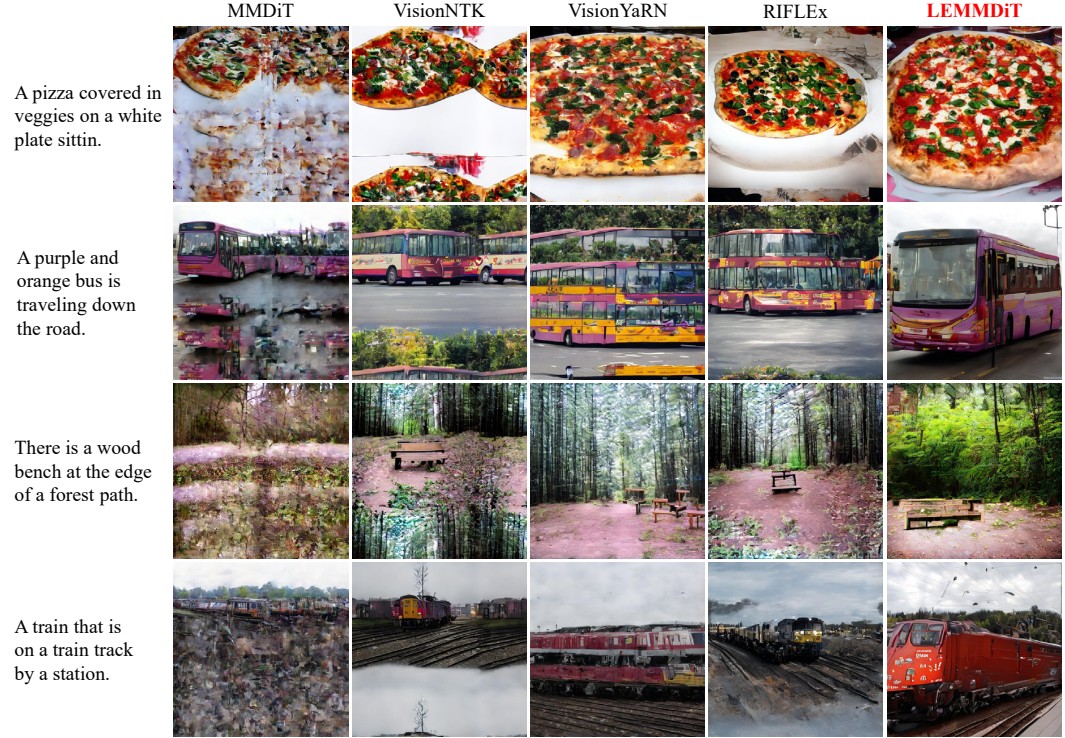

Figure 21: Qualitative comparison with other methods on text-to-image task. The prompts are located to the left of the image. We use the model trained on 256×256 COCO to generate images at 512×512. We set CFG=6.0. Best viewed when zoomed in.

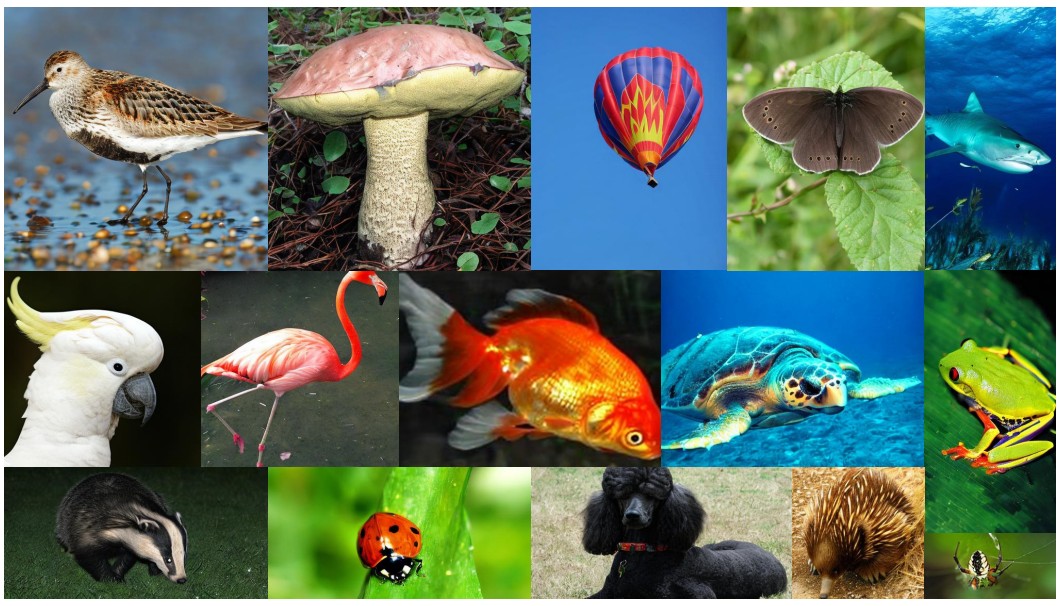

Figure 22: More arbitrary-resolution samples ($512^2$, 512×384, 384×512, 512×256, 256×512, $384^2$, $256^2$, 128×256). Generated from our LEDiT-XL/2 trained on ImageNet 256×256 resolution. We set CFG = 4.0.

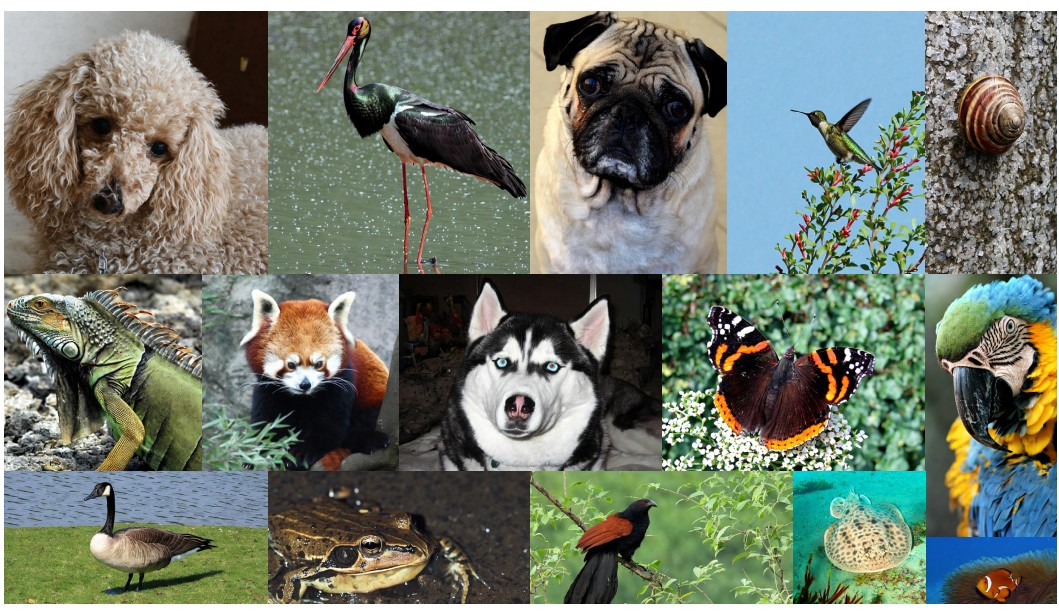

Figure 23: More arbitrary-resolution samples ($1024^2$, $1024\times768$, $768\times1024$, $1024\times512$, $512\times1024$, $768^2$, $512^2$, $256\times512$). Generated from our LEDiT-XL/2 trained on ImageNet $512\times512$ resolution. We set CFG = 4.0.

