# OpenReview forum: "LEDiT:  Your Length-Extrapolatable Diffusion Transformer without Positional Encoding"
_NeurIPS.cc/2025/Conference — NeurIPS 2025 poster_

### Official Review · Reviewer_f3XA · 2025-06-04

**Clarity:** 4
**Significance:** 3
**Originality:** 2
**Rating:** 5
**Confidence:** 5

**Summary:**

This paper introduces LEDiT, which improves the ability of diffusion transformers (DiTs) to generate larger images at test time. The method modifies DiT in two ways:

1. A causal/directional mask in self-attention, which prevents attention to below _and_ to the right of the query token. This mask encodes position information since every token has a different set of visible tokens for attention, thus distinguishing the 2D positions of tokens along the flattened sequence dimension.
2. A (second) convolution after the initial tokenization/convolution operation, while _not_ adding position embeddings to tokens.

The paper demonstrates that LEDiT generates higher quality images when extrapolating to larger images than baseline position encoding methods invented for both ViTs and DiTs.

**Questions:**

None

**Ethical Concerns:**

["NO or VERY MINOR ethics concerns only"]

**Final Justification:**

This paper introduces a novel position encoding method that improves image-size extrapolation for DiTs. It outperforms SOTA position encoding methods (most of which were invented for ViTs).

During the rebuttal period, the authors demonstrated that LEDiT outperforms LookHere (NeurIPS 2024) on image generation fine-tuning. In light of LookHere, the LEDiT method is not as novel as the paper claims as LEDiT is _not_ the first to introduce causal attention masks for ViTs for extrapolation. That being said, the method as a whole is novel and the empirical results are important, thus I recommend the paper is accepted.

**Limitations:**

Some limitations are briefly discussed.

**Quality:**

3

**Strengths And Weaknesses:**

Strengths:
- (I believe) this paper benchmarks many position encoding methods for image size extrapolation for DiTs. The failure cases, e.g. sin-cos embeddings, provide useful information. I don't believe previous DiT papers extrapolate from 224 to 1024 px.
- The empirical results are strong. LEDiT beats VisionYaRN (the 2nd best method) by 17 FID at 1024 px.
- The ablations are nice. They show that NoPE alone fails, causal self-attention _or_ the extra convolution performs poorly, but causal self-attention _with_ the extra convolution extrapolates well.
- The exact causal 2D attention mask is novel (see the discussion on this novelty below)
- The multi-dilation strategy during training is novel to ViTs/DiTs and seems to help extrapolation a bit

Minor weaknesses:
- LEDiT is slightly worse than prior methods in-distribution (i.e., when training image size equals testing image size) (Table 5 in the appendix)

Major weaknesses:
- LookHere (NeurIPS 2024: https://openreview.net/forum?id=o7DOGbZeyP) is not cited nor benchmarked. I will compare LookHere to LEDiT so that the authors can respond and we can discuss. Both papers introduce position encoding methods for vision transformers with the primary aim of improving image-size extrapolation. Both papers leverage 2D causal attention masks to limit the receptive field of each token, which is found to facilitate image-size extrapolation. LookHere introduced directional attention masks for ViTs, which _are_ 2D causal attention masks. The LookHere paper presents many mask variants, such as "only attend above", which LEDiT visualizes in Fig 6A. LEDiT proposes mask variants that did not appear in LookHere, for example LEDiT's _chosen_ variant called "mask lower right" (Fig 6D) is the complement of LookHere's "attend lower right" mask. The LEDiT submission presents 2D causal attention masks as the key contribution, thus the omission of LookHere in baselines and discussions is critical.

Key differences: LookHere combines 2D attention masks with ALiBi-style distance encoding. LEDiT combines 2D attention masks with a convolutional stem. In both papers, these additional components are critical — which is interesting! LEDiT uses the same attention mask for all heads, LookHere masks heads differently. LEDiT introduces a multi-dilation strategy, which seems to help. LookHere was demonstrated on image recognition tasks — and LookHere is directly useable in DiTs (just like the other _included_ baselines, RoPE, sin-cos, Yarn, etc.). LEDiT is demonstrated on image generation tasks.

---

> ### Author Rebuttal · Authors · 2025-07-30
>
> We thank the reviewer for their positive feedback and are pleased that they appreciate our comprehensive comparison of positional encoding methods for extrapolation in DiTs, the strong empirical results, and the thorough ablation studies, the design of LEDiT framework.
>
> >[W1]: The omission of LookHere in baselines and discussions is critical.
>
> Thank you for your valuable comments and constructive feedback. We agree there are some conceptual similarities between LEDiT and LookHere.  Both LEDiT and LookHere explore causal attention to enhance length extrapolation. LookHere carefully designs various combinations of causal masks to provide directional inductive biases. It introduces AliBi to penalize attention scores and demonstrates extrapolation improvement. LookHere conduct extensive experiments to demonstrate the extrapolation capabilities in classification tasks.
>
> However, **we clarify that there are key differences. Notably, directly adapting LookHere to image generation tasks does not yield effective length extrapolation**. We have provided a comprehensive comparison between LEDiT and LookHere in both methodology and performance.
>
> **The key differences between LEDiT and LookHere are summarized in the following table**:
> | Method      | Theoretical Contribution | Complementary Strategy to Causal Attention                     | Receptive Field   | Is Suitable for Extrapolation in Image Generation    |
> |-----------|----------|-------------------------------|------------|-------------|
> | LookHere     | No       | ALiBi | Restricted local  receptive field  | No    |
> | LEDiT  | Yes    | Multi-dilation Convolution                  | Global  receptive field      | Yes       |
>
> - **Theoretical Contribution:** We provide a rigorous theoretical framework that explains why causal attention is capable of encoding positional information and enabling length extrapolation. This not only facilitates effective image generation with robust extrapolation capabilities, but also offers valuable insights into the underlying mechanisms of length extrapolation. In contrast, LookHere does not investigate the reasons behind causal attention's ability to encode positional information; it merely states that attention with 2D masks can "limit the distribution shift that attention heads face when extrapolating" (Page 2).
>
> - **Complementary Strategy to Causal Attention & Receptive Field**: LEDiT not modify the causal attention mechanism itself. Instead, we introduce multi-dilation convolution as a complementary module to causal attention, **ensuring that the global receptive field of causal attention remains unaffected**. LookHere incorporates AliBi into causal attention. AliBi can be interpreted as a form of soft window attention, where an attention score penalty increases with distance, inherently restricts the receptive field of causal attention—even when LookHere explores different combinations of causal masks to cover the entire image.
>
> - **Is Suitable for Extrapolation in Image Generation**:
> we conduct detailed experiments and observe that LookHere fails to achieve effective length extrapolation in image generation tasks— whereas LEDiT demonstrates robust and reliable extrapolation capabilities.
>
>
> **Detailed Experimental Comparison**:
>
> We evaluate all LookHere variants (LH-180, LH-90, LH-45) and compare their performance with LEDiT. We fine-tune all models for 100K steps on ImageNet-256x256 by loading the DiT-XL/2 pretrained weights. Since AliBi is a key component of LookHere, we also report results for AliBi to facilitate a comprehensive discussion. As shown in the table below, **LEDiT outperforms both LookHere and AliBi at the training resolution as well as at higher resolutions**. We observe that, when extrapolated to higher resolutions, samples generated by LookHere preserve fine image details but suffer from severe object duplication. Since we are unable to provide visualization of generated images, we focus on quantitative comparisons using FID and precision metrics, which primarily reflect the structural accuracy and quality of the generated images. LookHere achieves lower FID and precision scores compared to LEDiT.
>
> | Method         | 512x512 FID↓   |  sFID↓   |  IS↑     |  Precision↑ |  Recall↑ | 256x256 FID↓ | sFID↓ | IS↑ | Precision↑ | Recall↑ |
> |------------------------|--------|---------|------------|---------|----------|---------|----------------|-------------|-------------|-------------|
> | AliBi            |   241.38  | 123.21     |  6.26   |   0.05  | 0.16   |  109.70  |  35.14          |  14.28      | 0.21 | 0.68|
> | LH-45            |  75.66   |   81.28   |   70.13  |  0.33   |  0.37  | 2.77  |   5.08         |   236.65     | 0.81|0.57
> | LH-90            |  80.84   |   84.13   |  67.74  | 0.34   | 0.37   |  2.72 |    4.93        |   242.92     | 0.82| 0.56
> | LH-180            |   66.93  |   83.09   |   82.39  |   0.39  | 0.36   | 2.54  |   4.94         |  248.47      |0.82 | 0.57
> | **LEDiT**     |  **35.86**    |   **67.97**     |    **139.91**       |  **0.52**      | **0.51**    | **2.38** |   **4.58**        |   **268.66**     | **0.83** | **0.58**|
>
> Table 1: Out-of-Distribution (512x512) and In-distribution (256x256) comparison between LEDiT and other methods. we report FID-10K for 512x512 and report FID-50K for 256x256
>
> This raises an interesting question: **why LookHere demonstrated on image recognition tasks but fails on image generation tasks?**  We present some insights:
>
> - **Consistent Higher Training Loss for LookHere:**  We report the mean loss over the last 1,000 training steps, as shown in the table below. On the training set, LookHere shows a higher loss compared to LEDiT and Sin/Cos PE, indicating that LookHere is relatively underfitting. Increasing the training steps to 150K does not further reduce the training loss, suggesting that the model has already converged. **The persistently high training loss suggests that the LookHere architecture may not be directly applicable to the image generation task**.
>
> | Method         | Training Loss   |
> |------------------------|--------|
> | **Sin/Cos PE (100K)**            |   **0.14005**  |
> | AliBi (100K)            | 0.14557    |
> | LH-45  (100K)          |  0.14301   |
> | LH-45  (150K)          |  0.14296   |
> | LH-90  (100K)         |   0.14283  |
> | LH-90 (150K)           | 0.14275    |
> | LH-180 (100K)           | 0.14134    |
> | LH-180 (150K)           | 0.14150    |
> | **LEDiT (100K)**           |   **0.14018**  |
>
> - **Image generation requires more precise positional information than image classification.** In image classification, the input is a clear image, and the model can rely on strong image content priors to make predictions. In contrast, image generation starts from pure noise, providing no prior information. Therefore, stronger positional encoding is essential for generation tasks.
>
> - **LookHere uses causal attention and AliBi, but these mechanisms are unable to encode precise positional information.** As discussed in our main paper, causal attention cannot accurately encode positional information. We also observed that AliBi performs poorly on in-distribution settings, indicating that it does not provide accurate positional information. We believe that while the combination of causal attention and AliBi works for image classification, their ability to capture positional information is limited, which makes LookHere unsuitable for image generation.
>
>
> - **The limited local receptive field of LookHere restricts its extrapolation capability.** A global receptive field is crucial for enabling the network to capture global context, which is essential for robust length extrapolation. The causal attention in LEDiT provides access to the entire global context, resulting in strong extrapolation ability. Although LookHere uses directional attention to cover the whole image, it penalizes long-range attention scores, which impairs its ability to model long-distance dependencies. Consequently, LookHere tends to generate repeated objects and exhibits inferior performance during extrapolation.
>
> In contrast, LEDiT combines simple causal attention with multi-dilation convolutions, achieving favorable performance in training resolution settings and demonstrating superior extrapolation ability. **Compared to LookHere, LEDiT exhibits better positional encoding capabilities in image generation tasks**. Thanks for your insightful comment. We will add the discussion and performance comparison with LookHere in the revised version.
>
>
> >[W2]: Slightly worse than prior methods in-distribution.
>
> Thank you for your insightful comment. **We have found that, by appropriately balancing causal attention and self-attention, our method achieves comparable in-distribution performance to RoPE while significantly outperforming existing extrapolation methods**. We conducted controlled ablation studies by varying the number of causal attention layers in LEDiT. With only one causal layer, LEDiT significantly outperforms all previous methods at higher resolutions. At the training resolution, LEDiT achieves performance comparable to the widely used RoPE, **demonstrating strong in-distribution performance while exhibiting superior extrapolation capabilities**. Due to space limitations, we respectfully refer you to our response to Reviewer xXsT's first question for further details.

---

> > ### Comment · Reviewer_f3XA · 2025-08-02
> >
> > Based on these new experiments, I will raise my score from a 3 (weak reject) to a 5 (accept). And I expect (and will double check) that the authors incorporate these comparisons and modify the novelty claims in the next iteration of the paper w.r.t. LookHere. Overall, the paper provides important empirical results to an understudied challenge, i.e., image size extrapolation.
> >
> > I have a few other comments for completeness and to help inform future readers of these reviews.
> >
> > > LookHere uses causal attention and AliBi, but these mechanisms are unable to encode precise positional information.
> >
> > This is speculative and not true IMHO. One way to test this could be to probe intermediate tokens for positional information.
> >
> > > The limited local receptive field of LookHere restricts its extrapolation capability
> >
> > LookHere does not have a "local receptive field". It does have a local bias, but so does LEDiT (through convolutions).
> >
> > > We fine-tune all models for 100K steps on ImageNet-256x256 by loading the DiT-XL/2 pre-trained weights.
> >
> > This is an important detail. LookHere was only demonstrated on training from scratch. Maybe LEDiT is better suited to this setting — which is a realistic setting, i.e., fine-tuning a model pre-trained with different position encoding.

---

> > > ### Author Response · Authors · 2025-08-07
> > >
> > > Dear Reviewer f3XA,
> > >
> > > We are happy to hear that our rebuttal addressed your concerns well. Also, we appreciate your support for our work. Following your insightful suggestions, we will incorporate these comparisons and update the modified novelty claims results in the revised version w.r.t. LookHere. We also appreciate your recognition of the practical value of the fine-tuning scenario (as many powerful models are based on DiT-pretrained weights) and the effectiveness of LEDiT in this context.
> > >
> > > We agree that LookHere's elegant design was demonstrated in the training-from-scratch setting. In our main paper, we also validated the effectiveness of LEDiT under the same scenario. Furthermore, we compared LookHere-180 and LEDiT when trained from scratch. However, training from scratch is time-consuming. Due to time and resource constraints, we have currently trained LookHere from scratch for up to 250K steps. The table below presents the performance at 250K steps on both in-distribution (256×256) and out-of-distribution (512×512) settings.
> > >
> > > | Method        | Out-of-Distribution 512x512 FID↓ | In-Distribution 256x256 FID↓|
> > > |------------------------|--------|---------|
> > > | Sin/Cos PE            |  241.87   | 27.91   |
> > > | LH-180            | 79.58  | 26.85  |
> > > | LEDiT     |  74.61   |   26.98    |
> > >
> > > Table 1: Comparison when trained from scratch. We report FID-10K for both out-of-distribution (512x512) and in-distribution (256x256) settings.
> > >
> > > **In the in-distribution scenario**, LookHere and LEDiT achieve comparable performance, both slightly outperforming Sin/Cos PE. This demonstrates that both methods can encode precise positional information. In the revised version, we will correct the statement in LookHere regarding the ability to encode accurate positional information. **In the out-of-distribution setting**, LEDiT outperforms LookHere. Visualizations of the generated samples indicate that LEDiT can better mitigates the issue of object duplication. We will continue training and provide a comprehensive comparison between LookHere and LEDiT, including additional visualizations, in the revised version. Additionally, we will revise the term "local receptive field" to "local bias" as recommended.
> > >
> > > Thank you once again for your invaluable feedback.
> > >
> > > Best regards,
> > >
> > > Authors

---

### Official Review · Reviewer_x8vf · 2025-06-25

**Clarity:** 3
**Significance:** 1
**Originality:** 1
**Rating:** 3
**Confidence:** 5

**Summary:**

This paper introduces LEDiT (Length-Extrapolatable Diffusion Transformer), a novel architecture that overcomes the resolution extrapolation limitations of traditional Diffusion Transformers (DiTs). Standard DiTs rely on explicit positional encodings (PEs) like RoPE, which degrade performance when generating images beyond training resolutions due to positional index distribution shifts. LEDiT replaces explicit PEs with causal attention, which implicitly encodes global positional information by leveraging variance differences in attention outputs. The theorem proves that causal attention’s output variance is inversely proportional to position, enabling implicit ordering. A locality enhancement module (multi-dilation convolution) is added to capture fine-grained details, complementing causal attention’s coarse-grained global information.
Experiments on ImageNet and COCO show LEDiT supports up to 4× resolution scaling (e.g., 256×256 to 512×512) with superior image quality. It outperforms state-of-the-art methods in FID, IS, and other metrics, even for arbitrary aspect ratios (e.g., 512×384) without multi-aspect-ratio training. Fine-tuning from pretrained DiTs for 100K steps suffices for strong extrapolation, highlighting efficiency.

**Questions:**

Please see weakness.

This work is very interesting as it can support length-extrapolation without additional training on these higher-resolution image data. However, LEDiT has not achieved SOT on either 256 or 512 resolutions. The practical significance of this work is not clear. I am glad to raise my score if the authors satisfactorily solve my concerns.

**Ethical Concerns:**

["NO or VERY MINOR ethics concerns only"]

**Final Justification:**

This paper introduces an interesting approach to improve the extrapolation. It achieves SOTA extrapolation performance compared with baselines.

During rebuttal, the authors provided more experiments to reveal why this method can work. I think the ablations on convolution kernel size and positional index regression could further improve the quality of this work.
However, the model does not achieve SOTA performance on either ImageNet-256 or ImageNet-512 benchmark, limiting the significance and practical applications of this work. Therefore, I will raise my score from 2 to 3 only.
I suggest the authors incorporate these experiments into the revised version.

**Limitations:**

The practical significance of this work is not clear.

**Quality:**

2

**Strengths And Weaknesses:**

# Strengths
1. Stronger Extrapolation Performance: Achieves 4× resolution scaling with better structural fidelity and details than RoPE-based methods (e.g., FiT, RIFLEx). Generates arbitrary aspect ratios without specialized training, demonstrating generalizability.
2. Theoretical Rigor: Causal attention implicitly encodes positional info, validated by theoretical proof of variance-position relationship. Theorem 3.1 mathematically links causal attention output variance to positional ordering, providing a foundation for implicit PE.

# Weakness
1. The contribution and significance of this paper are rather vague, because LEDiT has not achieved SOT on both 256 and 512 resolutions, especially at 512, where there are already many powerful models. The practical significance of this work is not clear.
2. Why Conv can provide positional information? Does kernel size and layer number affect the positional information?
3. To prove that causal attention can implicitly encode positional information, the author should provide the positional index regression experiments following LLM. The variance distribution in Figure 4 is not enough.
4. I observe slightly inferior performance on ImageNet-256x256. Causal attention’s reduced computational complexity may slightly degrade in-distribution performance (e.g., 256×256 FID slightly higher than DiT with RoPE). Does causal attention affect the semantic information captured in LEDiT?
5. Multi-dilation has been widely used in previous vision transformer literature[1,2], limiting the novelty of this work.

[1] Tokens-to-Token ViT: Training Vision Transformers from Scratch on ImageNet.
[2] Vision Transformer with Progressive Sampling.

---

> ### Author Rebuttal · Authors · 2025-07-31
>
> We thank the reviewer for the thoughtful feedback and are pleased that the reviewer appreciates the theoretical rigor in implicit positional encoding, and superior performance over state-of-the-art methods.
>
> However, we respectfully clarify that the concern raised is based on a misunderstanding of our paper. We have addressed and clarified all relevant points in the following response.
>
> >[W1]: The contribution and significance of this paper are rather vague. LEDiT has not achieved SOTA performance.
>
> Let us first clarify the significance of our work: **Our main goal is to improve the generative performance of DiT at resolutions higher than their training resolutions**, addressing a challenging and impactful out-of-distribution scenario relevant to many real-world applications, such as high-resolution medical imaging and high-definition film production. Current powerful DiT models, such as SD3, FLUX still struggle to generate images beyond the training resolution.
>
> To address this limitation. We introduce **a novel Diffusion Transformer without positional encoding (LEDiT)** for higher-resolution image generation, departing from the commonly used RoPE-based methods. **We provide both theoretical and empirical evidence that causal attention—the core component of LEDiT—implicitly encodes positional information, an that facilitates length extrapolation**.
>
> We conduct extensive experiments on both class-conditional and text-to-image generation tasks, demonstrating that LEDiT achieves superior extrapolation performance and outperforms state-of-the-art methods. For example, when extrapolating to 512×512 resolution, LEDiT reduces the previous best FID from 49.86 to 33.25. At 1024×1024 resolution, LEDiT surpasses VisionYaRN (the previous best method) by 17 FID points. **We believe our method achieves state-of-the-art extrapolation performance**.
>
> It is worth mentioning that the significance of our findings has also been acknowledged by other reviewers, who recognized that our work provide insightful theoritial analysis (Reviewer xXsT, JebD
> ) and shows compelling improvements in image fidelity (Reviewer xXsT
> , JebD, f3XA). It was also highlighted by reviewer 3ixD, who considered it quite important to the research community.
>
>
> >[W2]: Why Conv can provide positional information? Does kernel size and layer number affect the positional information?
>
> Previous works [1][2] have demonstrated that convolution with zero padding can leak local positional information, as cited in our main paper. The kernel size does not affect the leakage of positional information. As shown in the table below, the extrapolation performance to 512x512 resolution remains nearly identical across different kernel sizes.
>
> | 512x512  | FID↓ | sFID↓ | IS↑ | Precision↑ | Recall↑ |
> |------------------------|--------|---------|------------|---------|------------|
> | Kernel=(3,3)          |   35.86   |    67.97  |  139.91     |   0.52  |    0.51  |
> | Kernel=(5,5)          | 36.46  |     64.56   |     140.70   |    0.52    |    0.51  |
>
> When it comes to layer number, we conducted experiments by incorporating convolutional layers at different depths of LEDiT and found that adding convolutions to the early layers yields better performance than adding them to the deeper layers. We hypothesize that leveraging local positional information extracted by convolutions at earlier stages is more beneficial for diffusion transformer to leakage positional information, thereby leading to improved overall performance.
>
> | 512x512  | FID↓ | sFID↓ | IS↑ | Precision↑ | Recall↑ |
> |------------------------|--------|---------|------------|---------|------------|
> | Layer=1         |   35.86   |    67.97  |  139.91     |   0.52  |    0.51  |
> | Layer=10          | 49.27  |    79.52   |    93.47   |    0.43   |    0.46  |
>
> [1] How much position information do convolutional neural networks encode? ICLR 2020.
>
> [2] Segformer: Simple and efficient design for semantic segmentation with transformers. NeurIPS 2021.
>
>
> >[W3]: Provide the positional index regression experiments to prove that causal attention can implicitly encode positional information
>
> Thank you for your valuable comments. **The positional index regression experiment further verifies that causal attention can encode positional information**. Specifically, we trained an MLP to predict the position index of each token using the outputs of causal attention from a well-trained DiT as input. We conducted experiments on both 1D and 2D position regression tasks on ImageNet-256x256 to validate the effectiveness.
>
> For 1D causal attention, the MLP predicts the 1D position index of each token (e.g., {1}, {2}, ... {256}). For 2D causal attention, the MLP predicts the 2D position index (e.g., {1,1}, {1,2},...,{16,16}) for each token. The MLP is trained using L2 loss.
>
> We performed these experiments using DiT with 1D causal attention (variant (a) in the main paper) and DiT with 2D causal attention (variant (d)), and compared the results with DiT-NoPE, which cannot encode positional information. If the causal attention variants outperform DiT-NoPE, it demonstrates that the outputs of causal attention contain implicit positional information.
>
> During inference, we generated images using both DiT with causal attention and DiT-NoPE. At each of the 250 denoising steps, the MLP predicts the positional index from the features output by causal attention. We generated 100 images, resulting in 2,500 tests in total. We report the L2 loss between the predicted and ground truth position indices. Since the MLP is trained to predict positional indices, it cannot generalize to unseen positional indices when extrapolating to higher resolutions. Therefore, we perform inference with the MLP at the training resolution. Nevertheless, the significant performance gap compared to NoPE provides strong evidence that causal attention can implicitly encode positional information. As shown in the table below, **causal attention demonstrates a significant advantage over NoPE in both training loss and test error. This indicates that the position regressor can effectively learn positional information from the outputs of causal attention, providing further evidence that causal attention can implicitly encode positional information**.
>
> |    | Training Loss | Test Error |
> |------------------------|--------|---------|
> | 1D position regression          |     |      |
> | DiT-NoPE         | 5091.24  |     5265.85   |
> | **DiT with causal attention**         |  **97.42** |  **112.08**    |
> | 2D position regression         |     |      |
> | DiT-NoPE         | 12.57    |   13.30   |
> | **DiT with causal attention**        |  **1.20** |   **1.36**   |
>
>
> >[W4]: Multi-dilation has been widely used in previous vision transformer literature[1,2], limiting the novelty of this work.
>
> **We believe T2T-ViT [1] and PS-ViT [2] do not mention or utilize dilation-related concepts or techniques**. The multi-dilation training strategy is only one of the contributions presented in our paper. Our main contribution is improving the generative performance of DiT at resolutions higher than those seen during training. We introduce a novel Diffusion Transformer without positional encoding (LEDiT) and provide rigorous theoretical analysis as well as extensive experimental results to demonstrate the extrapolation capability of LEDiT. We believe our contributions are significant in the domain of length extrapolation.

---

> > ### Author Response · Authors · 2025-08-01
> > **Follow-up Clarification on Previous Response**
> >
> > Thank you again for the valuable feedback. We apologize for not directly answering the concern about "slightly inferior in-distribution performance" in our initial response. We would like to clarify that this issue has been thoroughly discussed, with detailed explanations and extensive supporting experiments, in our responses to Reviewer  xXsT ([W1&W3&Q2]) and Reviewer f3XA ([W2]). We kindly refer you to those responses for a comprehensive answer. We apologize for this oversight and appreciate your understanding. We hope that our previous responses address your concern as well.

---

> > ### Comment · Reviewer_x8vf · 2025-08-03
> >
> > Based on these new experiments, I will raise my score. I think the ablations on convolution kernel size and positional index regression could further improve the quality of this work. I suggest the authors incorporate these experiments into the revised version. Although it is not SOTA on standard benchmarks, it achieves SOTA extrapolation performance.

---

> > > ### Author Response · Authors · 2025-08-07
> > >
> > > Dear Reviewer x8vf,
> > >
> > > We are happy to hear that our rebuttal addressed your concerns well. Also, we appreciate your support for our work. Following your insightful suggestions, we will incorporate the ablations on convolution kernel size and positional index regression experiments in the revised version.
> > >
> > > Thank you once again for your invaluable feedback.
> > >
> > > Best regards,
> > >
> > > Authors

---

### Official Review · Reviewer_JebD · 2025-07-02

**Clarity:** 3
**Significance:** 3
**Originality:** 2
**Rating:** 5
**Confidence:** 4

**Summary:**

The paper demonstrates that causal attention with no positional encoding can efficiently encode global position information and proposes LEDiT architecture to efficiently extrapolate to resolutions higher than training resolution in diffusion models.

**Questions:**

The authors should include a baseline comparison with learnable position embeddings and analyze the trend in Figure 4 and Appendix B for the default “Mask Lower-right Corner” raster scan. Further, the authors should clarify if NoPE can effectively encode position information at higher resolutions. Finally, justification for the claim that attention scores are independent of inputs in (Lines 453–454) should be provided. I am willing to increase my score if authors can clarify these concerns.

**Ethical Concerns:**

["NO or VERY MINOR ethics concerns only"]

**Final Justification:**

All of my concerns including additional learnable positing embedding baseline and assumption in proofs have been resolved. Hence, I have raised my score.

**Limitations:**

yes

**Quality:**

3

**Strengths And Weaknesses:**

**Strengths**

1. The paper demonstrates theoretically and experimentally that causal attention with no position encoding can encode global information by utilizing variance in outputs at different positions in the sequence.
2. The proposed approach leads to notable improvements over baselines. Further, authors provide a comprehensive ablation study of various design choices.
3. The paper is overall well-written and easy to follow.

**Weaknesses**

1. Missing baseline: The authors should also compare the proposed approach with learnable position embeddings.
2. Potential issue in scaling: The variance appears to plateau after approximately 150 steps in Figure 4 (left). This raises concerns that at higher resolutions, for example 1024×1024 with 4096 token generation, most tokens may exhibit insufficient variance to encode positional information effectively.
3. While the assumption that inputs to causal self-attention are Gaussian is reasonable in Theorem 3.1, the assumption that attention scores are mutually independent of the inputs (Lines 453–454) is unclear and lacks justification.
4. Missing experiment analyzing trend in Fig 4 and Appendix B for “Mask Lower-right Corner” raster scan which is the default order in the paper.

---

> ### Author Rebuttal · Authors · 2025-07-30
>
> We thank the reviewer for their positive feedback and are pleased that they found our paper well-written and easy to follow. We appreciate the recognition of theoretical and experimental demonstration. We are also encouraged that the reviewer acknowledges the notable improvements.
>
> >[W1]: Missing baseline: learnable position embeddings.
>
> Similar to ViT and Swin Transformer, we replace the positional encoding (PE) in DiT with learnable positional embeddings. For length extrapolation, we interpolate the learnable positional embeddings to higher resolutions to ensure compatibility. As shown in the table below, at the training resolution, Learnable PE and LEDiT exhibit nearly comparable performance. When extrapolating to 512×512, **we observe a significant drop for Learnable PE**. This is likely because the interpolated positional embeddings at new spatial locations are not seen during training, leading to degradation.
>
> | Method         | 512x512 FID↓   |  sFID↓   |  IS↑     |  Precision↑ |  Recall↑ | 256x256 FID↓ | sFID↓ | IS↑ | Precision↑ | Recall↑ |
> |------------------------|--------|---------|------------|---------|----------|---------|----------------|-------------|-------------|-------------|
> | Learnable PE       |   208.45    |   139.38     |    5.23      |   0.02     |   0.02  | 2.38 |     4.69    |  **275.05**      | 0.82|0.58|
> | **LEDiT**         |  **35.86**    |   **67.97**     |    **139.91**       |  **0.52**      | **0.51**    | 2.38 |   **4.58**        |   268.66     | **0.83** | 0.58|
>
> Table 1: We report FID-10K for 512x512 and FID-50K for 256x256
>
> >[W2]: Scaling to higher resolutions such as 1024x1024.
>
> We respectfully clarify that a comparison at 1024×1024 resolution is presented in Table 7 of the appendix, where our approach demonstrates a clear performance advantage. Please refer to the appendix for detailed results.
>
> As discussed in Section 3.3 of the main paper, we have observed that the variance between local tokens becomes indistinguishable at higher resolutions. To address this, we introduce Locality Enhancement, which improves LEDiT's ability to capture local positional information.
>
> Furthermore, we evaluate LEDiT trained on ImageNet-256×256 and extrapolated to 1024×1024 resolution (a 16× length extrapolation). As shown in the table below, LEDiT outperforms other methods. However, the high FID suggests that aggressive resolution extrapolation remains challenging and warrants further exploration.
>
> | 1024x1024  | FID↓ | sFID↓ | IS↑ | Precision↑ | Recall↑ |
> |------------------------|--------|---------|------------|---------|------------|
> | DiT-LearnablePE          |   284.07   | 230.15       |  2.41       |  0.08   | 0.01   |
> | DiT-Sin/Cos PE          |  281.57    |   240.17     |     2.16    |  0.01   |   0.02 |
> | DiT-VisionNTK          |  333.01    |  244.23      |     1.96    |   0.22  |  0.00  |
> | DiT-VisionYaRN          |   228.41   |   199.62     |   7.60      |  0.03   |  0.09  |
> | DiT-RIFLEx          |  335.30    |    214.46    |    4.86     |    0.01 |  0.12  |
> | FiTv2-VisionNTK          |   342.54   |     260.28   |     2.75    |    0.01 | 0.00   |
> | FiTv2-VisionYaRN          |  338.12    |    241.12    |     2.93    |   0.01  |   0.00 |
> | **LEDiT**         | **212.97**   |  **169.88**    |     **10.14**    |   **0.05**    |    **0.14**    |
>
>
> >[W3]: Justification of the assumption that attention scores and v are mutually independent.
>
> Thank you for your insightful comments. **We observe that the correlation between the attention matrix and the value vectors is low during the early stages of denoising**. This empirical observation supports the validity of independence assumption. As discussed in the main paper (lines 128–143), our analysis primarily focuses on the early stage, where we show that causal attention mainly encodes positional information.
>
> **Theoretical Justification:**  The input sequence $\mathbf{x} = [x_1,...,x_n]$ can be approximated as i.i.d. Gaussian noise in the early denoising stage, as acknowledged by the reviewer. Each {$x_i$} is an independent Gaussian vector. The queries and keys are computed as:
> $q_i = x_i W_q$, $k_j = x_j W_k$
> Since the {$x_i$} are i.i.d., the sets {$q_i$} are also i.i.d. after linear transformation, the same as {$k_j$}. The attention score is: $S_{ij} = q_i^T k_j$.
> Because the {$k_j$} are i.i.d., the set {$S_{ij}$} for $j=1,...n$ are identically distributed random variables. After applying softmax $A=Softmax(S)$, the attention matrix $A_{ij}$  approaches a uniform distribution, and $\mathbb{E}[A_{ij}] = 1/n$. From this perspective, the attention matrix and the value have low correlation: changes in the value vectors do not significantly affect the attention matrix, resulting in low correlation.
> We conduct a toy experiment with a sequence length of 256 and a head hidden dimension of 72, consistent with the DiT-XL/2 configuration. We randomly initialized $W_q$, $W_k$, $W_v$, generated random Gaussian noise $x$, and computed the correlation coefficient between the attention scores $S$ and the values $V$ over 1000 trials. The average correlation coefficient was 0.03, indicating very low correlation. This provides theoretical support for our assumption.
>
> **Empirical Justification:** In practice, we also observe several layers show low correlation between the attention matrix and the value vectors during the early denoising stages of a trained diffusion transformer. Specifically, we report the correlation coefficients between the attention matrix and the value vectors across different timesteps and layers (see Table below). As shown, the correlation remains low in the early stages of denoising. While the correlation gradually increases in later stages—where the independence assumption no longer holds—we have demonstrated in the main paper that causal attention primarily encodes positional information during the early denoising steps. Therefore, this does not affect the validity of our justification.
>
> | Timestep\Layer | 0 | 14 | 22 |
> | --- | --- | --- | --- |
> | 999 | 0.010 | 0.065 | 0.066 |
> | 887 | 0.036 | 0.081 | 0.012 |
> | 774 | 0.033 | 0.048 | 0.116 |
> | 710 | 0.077 | 0.077 | 0.129 |
> | 642 | 0.064 | 0.120 | 0.121 |
> | 582 | 0.101 | 0.143 | 0.223 |
> | 526 | 0.119 | 0.176 | 0.207 |
> | 449 | 0.156 | 0.199 | 0.192 |
> | 401 | 0.141 | 0.169 | 0.145 |
> | 337 | 0.133 | 0.137 | 0.163 |
> | 265 | 0.160 | 0.084 | 0.170 |
> | 249 | 0.174 | 0.122 | 0.161 |
> | 189 | 0.213 | 0.191 | 0.199 |
> | 124 | 0.293 | 0.273 | 0.234 |
> | 0 | 0.371 | 0.336 | 0.275 |
>
> >[W3]: Missing experiment analyzing trend in Fig 4 and Appendix B for “Mask Lower-right Corner” variant.
>
> Thank you for your valuable suggestions. **The variance distribution under the "Mask Lower-right Corner" order is consistent with our main findings.** As this is a 2D scan variant, **the variance is expected to decrease progressively along the height or width axis**. Due to rebuttal space constraints, we provide a single table each for in-distribution and out-of-distribution results. Additional visualizations will be included in the revised version. These results demonstrate that the 2D causal scan variants still exhibit the existence of causal attention in DiT that satisfies the conditions outlined in our theorem.
>
> Additionally, **we conducted a position regression experiment to further demonstrate that causal attention can implicitly encode positional information**. For more details, please refer to our response to [W3] of Reviewer x8vf if interested.
>
>
> Height/Width | 0 | 2 | 4 | 6 | 8 | 10 | 12 | 14 | 15 | Mean Variance (Height)
> |---|---|---|---|---|---|---|---|---|---|---|
> 0 | 62.568 | 61.406 | 61.808 | 61.052 | 61.107 | 58.746 | 55.701 | 52.502 | 51.281 | **59.128** |
> 2 | 62.025 | 61.291 | 61.751 | 61.715 | 60.529 | 58.479 | 55.815 | 52.701 | 50.657 | **58.878** |
> 4 | 62.141 | 61.784 | 60.561 | 59.886 | 59.104 | 57.109 | 54.493 | 52.129 | 50.380 | **58.047** |
> 6 | 62.046 | 61.590 | 60.715 | 58.857 | 58.439 | 56.513 | 54.888 | 51.386 | 50.176 | **57.554** |
> 8 | 60.435 | 59.905 | 59.168 | 58.561 | 57.429 | 55.103 | 53.934 | 52.292 | 50.549 | **56.853** |
> 10 | 59.075 | 57.629 | 57.258 | 56.201 | 55.485 | 54.238 | 52.989 | 51.017 | 50.692 | **55.301** |
> 12 | 55.547 | 55.541 | 54.981 | 54.173 | 54.771 | 53.725 | 51.427 | 51.155 | 50.773 | **53.746** |
> 14 | 53.556 | 52.497 | 52.061 | 51.996 | 53.142 | 51.323 | 51.104 | 51.472 | 50.741 | **52.047** |
> 15 | 50.885 | 50.879 | 50.614 | 50.155 | 50.687 | 50.706 | 50.610 | 51.098 | 50.415 | **50.715** |
> Mean Variance (Width) | **59.215** | **58.649** | **58.246** | **57.213** | **57.041** | **55.336** | **53.573** | **51.788** | **50.670** | - |
>
> Table 2: Timestep: 943, Layer: 20. In-distribution (256x256) settings.
>
>
> Height/Width | 0 | 4 | 8 | 12 | 16 | 20 | 24 | 28 | 31 | Mean Variance (Height)
> |---|---|---|---|---|---|---|---|---|---|---|
> 0 | 89.495 | 79.405 | 74.726 | 71.715 | 67.887 | 62.126 | 55.560 | 45.446 | 28.546 | **65.118** |
> 4 | 82.514 | 78.698 | 74.238 | 69.688 | 62.903 | 55.681 | 48.119 | 40.209 | 27.356 | **61.450** |
> 8 | 78.711 | 74.260 | 68.082 | 63.037 | 57.270 | 50.776 | 43.420 | 35.626 | 28.701 | **56.517** |
> 12 | 74.934 | 69.286 | 63.372 | 57.743 | 51.852 | 45.344 | 38.704 | 31.527 | 26.989 | **51.836** |
> 16 | 72.041 | 64.826 | 57.983 | 52.671 | 46.417 | 41.822 | 35.249 | 29.439 | 26.042 | **47.867** |
> 20 | 66.171 | 58.735 | 52.689 | 46.901 | 40.970 | 36.559 | 31.926 | 27.396 | 25.330 | **43.194** |
> 24 | 57.264 | 52.083 | 45.432 | 39.545 | 35.789 | 31.558 | 28.112 | 26.279 | 27.037 | **37.930** |
> 28 | 47.008 | 40.636 | 35.092 | 31.552 | 28.650 | 26.924 | 26.219 | 25.084 | 26.212 | **31.585** |
> 31 | 27.496 | 26.517 | 26.821 | 26.373 | 25.777 | 26.950 | 25.257 | 25.255 | 24.567 | **26.298** |
> Mean Variance (Width) | **67.905** | **62.102** | **56.820** | **51.894** | **46.846** | **41.982** | **37.188** | **31.573** | **26.563** | - |
>
> Table 3: Timestep: 750, Layer: 15. Out-of-distribution (512x512) settings.

---

> > ### Comment · Reviewer_JebD · 2025-08-06
> >
> > Thanks to authors for their detailed response. All of my concerns have been resolved and the manuscript should be updated to reflect additional baselines and clarifications. Hence, I have raised my score.

---

> > > ### Author Response · Authors · 2025-08-07
> > >
> > > Dear Reviewer JebD,
> > >
> > > We are happy to hear that our rebuttal addressed your concerns well. Also, we appreciate your support for our work. Following your insightful suggestions, we will update the additional baselines and clarifications in the revised manuscript.
> > >
> > > Thank you once again for your invaluable feedback.
> > >
> > > Best regards,
> > >
> > > Authors

---

### Official Review · Reviewer_xXsT · 2025-07-08

**Clarity:** 3
**Significance:** 3
**Originality:** 2
**Rating:** 5
**Confidence:** 3

**Summary:**

LEDiT introduces a novel No Positional Embedding (NoPE) Diffusion Transformer (DiT) for image generation, enabling models trained on low-resolution data to generate high-resolution images during inference. It overcomes the limitations of explicit positional encodings (PEs) by:

1. Implicit Positional Encoding: Leveraging an observed per-token variance property, LEDiT replaces self-attention with causal attention during specific timesteps. This implicitly encodes positional information, removing the need for explicit PEs.

2. Enhanced Locality: A single dilated convolution layer is added after patchification to capture fine-grained local features, ensuring a balance of global and localized information.

This combination allows LEDiT to achieve impressive length extrapolation and arbitrary resolution generation, a critical advancement for high-resolution image synthesis.  The authors show significant improvements over NOPE in baselines when applying a combination of causal attention + self-attention with a single dilated convolutional layer.

By combining these strategies, LEDiT enables DiTs trained on low-resolution data to generate high-resolution images during inference without loss of quality, making them more adaptable to diverse output requirements.

**Questions:**

1. The authors have also shown results on text to image tasks, however, it would be good to see how does the T2I alignment metric perform with the proposed architecture. It would be good to understand if the architecture would compromise semantic coherence for dense prompts ?

2. Could you elaborate on potential trade-offs of replacing self attention layers with causal attention's unidirectional flow.
. Is there any insight on why the recall drops ?

**Ethical Concerns:**

["NO or VERY MINOR ethics concerns only"]

**Final Justification:**

As mentioned in my review, the method is simple and elegant approach to context extrapolation without Positional Embedding.
The authors in the rebuttal have responded to two of my concerns
1.  Impact of causal attention on T2I alignment: Table on clip scores shows consistant improvement against existing PE baselines. The authors have also compared against baselines with learnable positional embeddings.
2. Increasing in the number of causal attention layers indicate improvements in the quality on default configuration

Due to extensive empirical investigation I have raised my score.

**Limitations:**

The paper is well motivated and well written. While the combination of causal attention with convolution has been applied in litterature, the strength of the paper lies in demonstrating that these building blocks could also be applied in context of length extrapolation without positional embeddings. The analytical insights into per-token variance are particularly helpful in strengthening the claim of novelty. However, I have a few concerns I have raised for which I give a borderline accept.

**Quality:**

3

**Strengths And Weaknesses:**

Strengths:

1. The paper offers valuable analytical and empirical insights into per-token variance, highlighting the temporal impact of positional information during diffusion. The causal attention's significant computational savings over full attention is a notable practical advantage.

2.  The combination of causal attention and a convolutional layer ("Enhanced Locality") demonstrably boosts performance over baseline DiTs without explicit PEs, effectively tackling length extrapolation.

3. LEDiT shows compelling improvements in image fidelity (FID, IS) when generating images at significantly higher resolutions than trained, confirming its primary claim of length extrapolability and arbitrary aspect ratio support.

Weakness:


1. LEDiT is a continuous diffusion model that uses a specific type of masked attention (causal attention) to achieve length extrapolation by implicitly encoding positional information. Most recent multi-modal generation models follow bi-directional attention increases contextual representation.

2. Training on smaller datasets (ImageNet, FFHQ, LSUN) rather than large-scale, diverse sets like LAION-5B raises concerns about LEDiT's generalizability and real-world applicability at scale.

3. The unidirectional nature of causal attention, compared to bidirectional attention in leading multimodal models, might limit the model's ability to capture full contextual representations, potentially reducing generative diversity (recall). This trade-off needs further analysis and discussion.

---

> ### Author Rebuttal · Authors · 2025-07-30
>
> We thank the reviewer for their positive feedback and are pleased that they found our paper to be well-motivated and well-written. We appreciate the recognition of our novel implicit positional encoding and the analytical insights provided, which enable robust length extrapolation and demonstrate substantial improvements over existing methods.
>
> >[W1&W3&Q2]: The trade-offs of replacing self-attention layers with causal attention's unidirectional flow needs further analysis and discussion. Whether causal attention might potentially reducing generative diversity (recall); The insight on why the recall drops.
>
> Thank you for your insightful comments. **We have found that, by appropriately balancing causal attention and self-attention, our method achieves in-distribution performance comparable to RoPE, while significantly outperforming existing methods on extrapolation tasks.**
> To provide a more comprehensive analysis, we conducted controlled ablation studies by varying the number of causal attention layers in LEDiT. Our results indicate that **increasing the number of causal layers enhances length generalization, with only a minor reduction in in-distribution performance. Compared to previous methods, our approach achieves a more favorable trade-off between these aspects.**
> Table 1 compares different LEDiT variants and other positional encoding methods under both in-distribution and out-of-distribution settings. Notably, with just a single causal layer, LEDiT significantly outperforms all previous methods at higher resolutions. At the training resolution, LEDiT matches the widely adopted RoPE in performance, **demonstrating that LEDiT preserves strong in-distribution results while providing superior extrapolation capability**. Increasing the number of causal layers further enhances extrapolation, with only a slight reduction in in-distribution performance. While the optimal trade-off may vary by application, our results indicate that the default configuration described in our paper (i.e., 14 causal layers) achieves an effective balance between in-distribution and out-of-distribution performance.
>
> | Method         | 512x512 FID↓   |  sFID↓   |  IS↑     |  Precision↑ |  Recall↑ | 256x256 FID↓ | sFID↓ | IS↑ | Precision↑ | Recall↑ |
> |------------------------|--------|---------|------------|---------|----------|---------|----------------|-------------|-------------|-------------|
> | Learnable PE       |   208.45    |   139.38     |    5.23      |   0.02     |   0.02  | 2.38 |     4.69    |  275.05      | 0.82|0.58|
> | Sin/Cos PE         |   215.69    |  180.75      |    8.09       |    0.05    |   0.15  | 2.27 |       4.60    |    278.24    |0.83|0.57
> | VisionNTK (RoPE-based)          |   251.85   |   153.40   |    8.73       |   0.04    | 0.26    | 2.33 |    4.58       |  272.02      |0.83|0.58
> | VisionYaRN (RoPE-based)          |   53.75   |   76.62    |    107.75      |   0.41    |   0.53  |  -  |    -        |       -  |- |-
> | RIFLEx (RoPE-based)             |   256.1    |   172.54    |    10.54      |   0.04    |  0.18   |  -  |    -        |       -  |- |-
> | **LEDiT (1 Causal layer)**   |   **37.70**    |   **65.20**    |    **137.57**       |   **0.52**    |  **0.52**   | **2.33** |   **4.53**        |  **273.84**      |**0.83**|**0.57**|
> | LEDiT (2 Causal layers)      |  38.60    |    68.92    |   134.46        |     0.50   |   0.51  | 2.35 |     4.61      |  268.69      | 0.83 | 0.57 |
> | LEDiT (4 Causal layers)      |   41.91   |   69.58     |    127.40       |  0.49      | 0.53    | 2.39 |   4.55        |   270.02     |0.83 | 0.57|
> | LEDiT (8 Causal layers)      |  39.87    |    69.18    |    133.82       |   0.51     |  0.51   | 2.34 |   4.57        |   267.69     | 0.83| 0.58|
> | **LEDiT (14 Causal layers)**     |  **35.86**    |   **67.97**     |    **139.91**       |  **0.52**      | **0.51**    | **2.38** |   **4.58**        |   **268.66**     | **0.83** | **0.58**|
> | **LEDiT (28 Causal layers)**       |  **32.61**    |   **61.06**     |    **162.58**       |   **0.57**     |  **0.51**   | **2.40** |   **4.65**        |   **264.81**     | **0.83**| **0.57**|
>
> Table 1: Out-of-Distribution (512x512) and In-distribution (256x256) comparison between LEDiT variants and other methods. we report FID-10K for 512x512 and report FID-50K for 256x256
>
> > Whether causal attention might potentially reducing generative diversity (recall); The insight on why the recall drops.
>
> **Causal attention is not the cause of recall reduction.** In fact, we emphasize that LEDiT even achieves slightly better recall than DiT in the in-distribution setting (see Table 1), indicating that causal attention does not compromise sample diversity.
>
> In the length extrapolation scenario, both Table 2 of the FiT paper [1] and Table 1 of our work consistently reveal two distinct method categories:
>
> - **Category I:** Methods with lower FID (e.g., LEDiT, DiT-VisionYaRN) tend to exhibit higher precision but lower recall.
> - **Category II:** Methods with higher FID (e.g., DiT-NTK) tend to exhibit lower precision but higher recall.
>
> For clarity, we reports the performance of various PE and LEDiT when extrapolated to 384×384 resolution, all trained on ImageNet-256×256.
>
> | 384x384  | FID↓ | Precision↑ | Recall↑ |
> |------------------------|--------|---------|------------|
> | VisionNTK (RoPE-based)          |  71.23   |   0.33  |    0.51    |
> | VisionYaRN (RoPE-based)          |  13.51  |    0.71 |     0.39     |
> | RIFLEx (RoPE-based)             |    57.88  |   0.37   |   0.55   |
> | LEDiT  |    9.34   |    0.78  |      0.39    |
>
> Category I methods generate high-quality samples with limited diversity, whereas Category II methods frequently produce a substantial proportion of broken images, as shown in the visualizations in our appendix. **We argue that Category I methods are preferable, consistent with the statement from the precision/recall paper[2] (page 5): *"It is unclear which application might favor setup D (Category II), where practically all images are broken, over setup B (Category I), that produces high-quality samples at a lower variation."***
>
> **The reduction in recall can be attributed to out-of-distribution (OOD) generalization challenges**. When models are trained on low-resolution data, the learned data distribution  $P_l$ does not fully capture the high-resolution manifold $P_h$. Consequently, when extrapolating to high-resolution, models tend to generate samples concentrated in the intersection region $P_e = P_l \cap P_h$. Samples from $P_e$ simultaneously satisfy both the higher-resolution requirements and the learned data distribution. However, since $P_e$ is a subset of $P_l$, this restriction reduces diversity, leading to lower recall.
>
> [1] FiT: Flexible Vision Transformer for Diffusion Model, ICML 2024.
>
> [2] Improved Precision and Recall Metric for Assessing Generative Models, NeurIPS 2019.
>
> >[W2]: Training LEDiT on large-scale, diverse sets like LAION-5B to test generalizability.
>
> Due to time constraints and limited computational resources,
> we currently face challenges in applying LEDiT to large-scale sets like LAION-5B.  Following previous works DiT, SiT, REPA, we conduct extensive experiments on class-conditional and text-to-image generation tasks on ImageNet and COCO dataets. We also believe that LEDiT can improve training on large-scale datasets, which contain images with diverse resolutions and aspect ratios. By removing the constraints of explicit positional encoding, LEDiT can better accommodate images of varying resolutions, which may lead to improved training outcomes.
> In future work, we plan to progressively extend LEDiT to larer-scale data to evaluate its generalization and scalability in more complex generative settings.
>
> >[Q1]: How does the T2I alignment metric perform with the proposed architecture. It would be good to understand if the architecture would compromise semantic coherence for dense prompts
>
> **LEDiT maintains semantic coherence for dense prompts.** We use 40,504 captions from the COCO dataset to generate images and compute the CLIP score as the text-to-image alignment metric. The comparison is presented in the following table. LEDiT consistently outperforms other methods in terms of semantic coherence.
>
> | 512x512  | CLIP Score↑ |
> |------------------------|--------|
> | MMMDiT        |  22.94  |
> | MMDiT-ViNTK         |  24.91  |
> | MMDiT-ViYaRN          |    23.37  |
> | MMDiT-RIFLEx          |    27.14  |
> | **LEMMDiT**  |    **27.82**   |

---

> > ### Comment · Reviewer_xXsT · 2025-08-08
> > **rebuttal response**
> >
> > Thank you for your response my concerns  on impact of causal attention  have been answered. In addition authors have shown the semantic coherence improvement on CLIP scores. I will increase my rating

---

> > > ### Author Response · Authors · 2025-08-08
> > >
> > > Dear Reviewer xXsT,
> > >
> > > We are happy to hear that our rebuttal addressed your concerns well. Also, we appreciate your support for our work. Following your insightful suggestions, we will update the corresponding experimental results in the revised version.
> > >
> > > Thank you once again for your invaluable feedback.
> > >
> > > Best regards,
> > >
> > > Authors

---

### Note · Authors · 2025-08-14

Dear Reviewers, AC, SAC, and PC,

We appreciate your valuable time in reviewing our manuscript.

In this work, we investigate the **underexplored challenge of length extrapolation in DiT**. We propose LEDiT, which removes explicit positional encodings (e.g., RoPE) and significantly enhances DiT's ability to extrapolate to untrained resolutions.

As reviewers noted, **LEDiT demonstrates impressive (xXsT, JebD, x8vf, f3XA) and comprehensive (JebD, f3XA) results, with helpful theoretical analysis (xXsT, JebD, x8vf). The manuscript is well-written and well-motivated (xXsT, JebD)**. Key concerns and our responses:

- **Trade-off between training and higher resolution.** Extensive ablation studies show that LEDiT can achieve comparable performance with existing methods (e.g. FID: LEDiT 2.33, RoPE 2.33) at the training resolution, while still demonstrating significant advantages when extrapolating to higher resolutions (e.g. FID: LEDiT 37.70, VisionYaRN 53.75).

- **Positional index regression.** Both 1D and 2D positional index regression experiments demonstrate that the positional information encoded by causal attention can be effectively extracted by neural networks.

- **Comparison with additional methods.** LEDiT outperforms Learnable PE and LookHere (e.g., FID: LEDiT 35.86, Learnable PE 208.45, LookHere 66.93).

- **Justification of the assumption.** The low correlation between the attention matrix and the value during the early denoising stages supports the assumption in the theorem.

- **Convolution kernel size.** The kernel size does not affect the extrapolation of LEDiT.

- **Scaling to aggressive extrapolation.** LEDiT maintains superior performance under aggressive extrapolation settings (e.g., FID: LEDiT 212.97, VisionYaRN 228.41).

Furthermore, We have trained LookHere-180 from scratch up to 400K steps. **LEDiT consistently shows better extrapolation and comparable in-distribution performance. Combined with our fine-tuning advantages over LookHere, these findings further highlight the effectiveness of LEDiT**.

|Method|512x512 FID↓|256x256 FID↓|
|-|-|-|
|Sin/Cos PE|242.17|21.46|
|LH-180|79.96|21.25|
|LEDiT|74.28|21.36|

Table 1: FID-10K w/o classifier-free guidance.

We have addressed all concerns and **appreciate the reviewers raising the score**. We believe that **the superior performance of LEDiT can advance the practicality of DiT, and offers new insights into transformer design**.

Thank you for your valuable time.

Best regards,

Authors

---

### Decision · Program_Chairs · 2025-09-17

**Decision:**

Accept (poster)

**Comment:**

The paper introduces LEDiT, a diffusion transformer architecture that aims to enable diffusion models to generalize to lengths beyond their training regime. LEDiT enables resolution extrapolation up to 4 times with improved image quality over state-of-the-art methods, including support for arbitrary aspect ratios without multi-aspect-ratio training. The reviewers originally raised various concerns, including the relationship with prior work that was not cited. I believe the authors have now become aware of this work and they will include it in the camera-ready version. Another concern was the scale of the experiments as raised by multiple reviewers. After the rebuttal the reviewers have been satisfied, while I do believe ImageNet is a reasonable dataset for the experiments. Therefore, this paper should be revised for the camera-ready but it presents an elegant solution.